# Vision-driven metasurfaces for perception enhancement

Tianshuo Qiu[1,2,3,8], Qiang An[1,8], Jianqi Wang[1] ✉, Jiafu Wang [4] ✉,
Cheng-Wei Qiu [5] ✉, Shiyong Li[6], Hao Lv[1] ✉, Ming Cai[2], Jianyi Wang[7], Lin Cong[1] &
Shaobo Qu[4] ✉

Metasurfaces have exhibited unprecedented degree of freedom in manipulating electromagnetic (EM) waves and thus provide fantastic front-end interfaces for smart systems. Here we show a framework for perception enhancement based on vision-driven metasurface. Human's eye movements are matched with microwave radiations to extend the humans' perception spectrum. By this means, our eyes can "sense" visual information and invisible microwave information. Several experimental demonstrations are given for specific implementations, including a physiological-signal-monitoring system, an "X-ray-glasses" system, a "glimpse-and-forget" tracking system and a speech reception system for deaf people. Both the simulation and experiment results verify evident advantages in perception enhancement effects and improving information acquisition efficiency. This framework can be readily integrated into healthcare systems to monitor physiological signals and to offer assistance for people with disabilities. This work provides an alternative framework for perception enhancement and may find wide applications in healthcare, wearable devices, search-and-rescue and others.

Human visual system[1–3], a remarkable network that allows us to receive light input into our eyes and to transform light information into colorful and rich experiences, is the most important pathway for humans to obtain information from the external world. Our eyes use visible light to obtain information about our surroundings, a process begins when the cornea and lens refract light from objects and surfaces in the world to form a panoramic, hemispheric image on the retina, a thin layer of nerve tissue that lines the inner surface of the eyeball. Like pixels in a digital camera, photoreceptor cells absorb the light arriving at the retina and photochemically transform the absorbed light energy into neural signals. Impulses from the photoreceptors are then transmitted to the visual cortex of the brain via a visual pathway. The visual system is an important part of the brain nerve center[4–6]. More than 80% of sensory information that our brain receives comes from the visual system, which provides us with a personal representation of our surrounding environment. The process resulting in vision is not only about the perception of the environment by capturing energy but also about its interpretation. The information obtained from visual system directly affects human life and thinking. Our daily tasks, including perception, decision making as well as action, are highly dependent on vision. Thus, research on visual systems has become a hot spot and the source of interactive scientific research in recent years. However, human eyes can only perceive visible light which occupies a very narrow frequency band in the entire EM spectrum. This restricts the visual system to obtain information from other frequency bands, resulting in a significant decrease in the ability to access information

[1]Department of Biomedical Engineering, Fourth Military Medical University, Xi'an, China. [2]Fundamentals Department, Air Force Engineering University, Xi'an, China. [3]State Key Laboratory of Millimeter Waves, Southeast University, Nanjing, China. [4]Aerospace metamaterials laboratory of SuZhou National Laboratory, Suzhou, China. [5]Department of Electrical and Computer Engineering, National University of Singapore, Singapore, Singapore. [6]School of Integrated Circuits and Electronics, Beijing Institute of Technology, Beijing, China. [7]Department of Neurology, the First Affiliated Hospital of Xi'an Jiaotong University, Xi'an, China. [8]These authors contributed equally: Tianshuo Qiu, Qiang An. ✉e-mail: fmmuwangjq@163.com; wangjiafu1981@126.com; chengwei.qiu@nus.edu.sg; fmmulvhao@fmmu.edu.cn; qushaobo@126.com

when the target is obstructed. Moreover, previous researches on visual search[7], spatial attention[8,9], and change blindness[10,11] have shown a limited ability of human observers to obtain and process all information present within a visual scene. Faced with a large amount of visual information, the visual system can only selectively handle a small part. When high-intensity work must be persistent for a long time or multiple tasks within short time intervals must be performed, this limitation may waste a lot of useful content and reduce the efficiency of information acquisition. Therefore, it is of great significance to figure out a method of making the most of visual information, which is expected to improve the efficiency of information acquisition in terms of achieving perceptual enhancement in multiple frequency bands of the EM spectrum.

Recently, metasurfaces have attracted great attention from engineers and researchers. Metasurfaces[12-20], two-dimensional artificial surfaces composed of sub-wavelength unit cells, have provided unprecedented degree of freedom in manipulating EM waves upon interfaces, including polarization rotation[21,22], vortex beam generation[23,24], amplitude and phase modulation[25,26]. Moreover, metasurfaces can cover almost all the frequency bands of the EM spectrum including microwave[27] and visible frequencies[28,29]. Due to high efficiency, low loss and small thickness, metasurface has been regarded as a popular notion and a promising candidate for powerful control over EM waves. Since 2014, N. Engheta and T. J. Cui proposed the concepts of digital metasurfaces[30] and coding metasurfaces[31] in order to balance complexity and simplicity in the development in science and engineering. On this basis, EM metamaterials have developed their own research paradigm. Many exotic functionalities can be realized such as cloak[32,33], computer vision[34,35], holograms[36], communications[37-39], brain waves[40,41] and others[42-45]. Under such considerations, we therefore associate human visual perception with microwave metasurfaces, with the expectation that metasurfaces improve human perception ability and make the most of visual information. To this end, microwave metasurfaces driven by vision activities, such as eye movements, should be firstly developed. As the window on mind and brain[46], many real-world visual perceptions are accompanied by characteristic eye movement behaviors. The scientific researches on eye movement have been conducted for more than 80 years and have widely used in advertising, reading, space cognition and so forth. The noninvasive, reliable, rapid, and simple measurement of eye movement offers the unique opportunity to gain deeper insights into the underlying mechanism of visual information processing[47-50]. Using the eye tracker[51-53], it is possible to record the gaze time, gaze position, and the interested area of the viewer, and thus to provide a bridge between metasurfaces and vision to enhance perception. This may result in a subversive breakthrough in the field of human vision, which enables visual perception in multiple frequency bands, multi-target tracking, penetrable vision, and improves the analysis efficiency of visual information. This is also in accordance with one of the important future development directions of metasurfaces, that is, biologically-driven intelligent metasurfaces. Moreover, such vision-driven metasurfaces may expand the human senses and provide healthcare for the disabled.

Inspired by this, in this work, we propose a framework for visual perception information enhancement system based on vision-driven metasurfaces. In our method, we connect the eye movements with microwave radiations of metasurfaces via eye tracker, where radiation beam is controlled automatically by eye movements. Visual information and invisible microwave information are synergized to extend the spectrum of humans' perception, allowing humans to perceive physiological signals, as well as the location and motion of hidden subjects. Afterward, a smart tracking system, which is capable of lightening the burden on the senses and improving the efficiency of information acquisition, is also proposed through the target selection of human eyes and tracking algorithm. At last, the visibility of speech signals is realized with the help of this metasurface system. In more details, the vibration driven by the sound wave is captured and converted into microwave information, and then into visual characters directly visible to human eyes to offer assistance for deaf people. Experimental verifications are presented to demonstrate the stronger information acquisition ability of the vision-driven metasurface framework, including the physiological-signal-monitoring system, X-ray-glasses system, glimpse-and-forget tracking system and barrier-free real-time speech reception system for deaf people. The simulated and measured results verify the design philosophy and the framework. Through a variety of sensing methods in multiple frequency bands, combined with signal processing, computer vision and other technical means, the system improves the human's ability to process visual information and compensates for the shortcomings of human vision. The introduction of metasurface enables humans with stronger information acquisition and perception ability. It can help humans obtain information in multiple frequency bands and realize perception under blocked conditions. This method can likewise help humans identify target of interest, assist target-tracking and decision-making. Moreover, the system breaks through the perception barriers of different organs, helps the deaf people obtain part of the hearing information with their eyes, realizing disabled healthcare. With the help of human visual perception ability, the proposed method allows the metasurface and its intelligent platform with more degree of freedom as well as more interesting functions. This work provides an alternative framework for perception enhancement and may find wide applications in healthcare, new-generation communication, wearable devices, search-and-rescue, target-tracking and others.

## Results

### Architecture of the perception enhancement system

Here, we propose a smart metasurface system architecture through a combination of multiple means, as outlined in Fig. 1(a). Eye tracking glasses, the wearable solution for capturing objective measures of cognitive workload and viewing behavior, are well-suited to providing reliable oculomotor data at the required spatial and temporal resolutions. When the user wears the eye tracker and runs the program, the relevant eye movement data including gaze point, eye saccades, blinks and so on can be extracted. On this basis, data processing program is used to process the raw data of eye movement. Through analysis of the area of interest (AOI) and the state of the subject (e.g. attention), areas that need to be tracked or perceptually enhanced are identified, which lays the foundation for control of programmable metasurface. It is worth mentioning that a wearable display attaches to the eye tracker in some demos, so that the observer can see the processed microwave band information while still looking at the target in front. Unlike the traditional means of displaying microwave information on a PC monitor, the extra screen on the eye tracker displays microwave information without letting the observer take eyes off the surrounding world.

Programmable metasurfaces are accompanied by digital coding characterization, in which the EM responses are manipulated by the digital-coding sequences. With the aid of field programmable gate array (FPGA) and digital-analog conversion (DAC) module, as the voltage applied to the diode changes, the digital-coding sequences change accordingly and thus radiation angle can be tuned. Therefore, in accordance with AOI and subjects' states, the algorithm can determine the reaction of the metasurface by itself and instruct the FPGA to change the programmable metasurface configuration and then digital-coding sequences are determined. In this way, significantly distinct functions of programmable metasurface can be adjusted to facilitate good-shaped directional radiation beams and other smart features. Moreover, by transparent dielectric substrates and metal mesh patterns, the proposed metasurface structure can achieve high visible

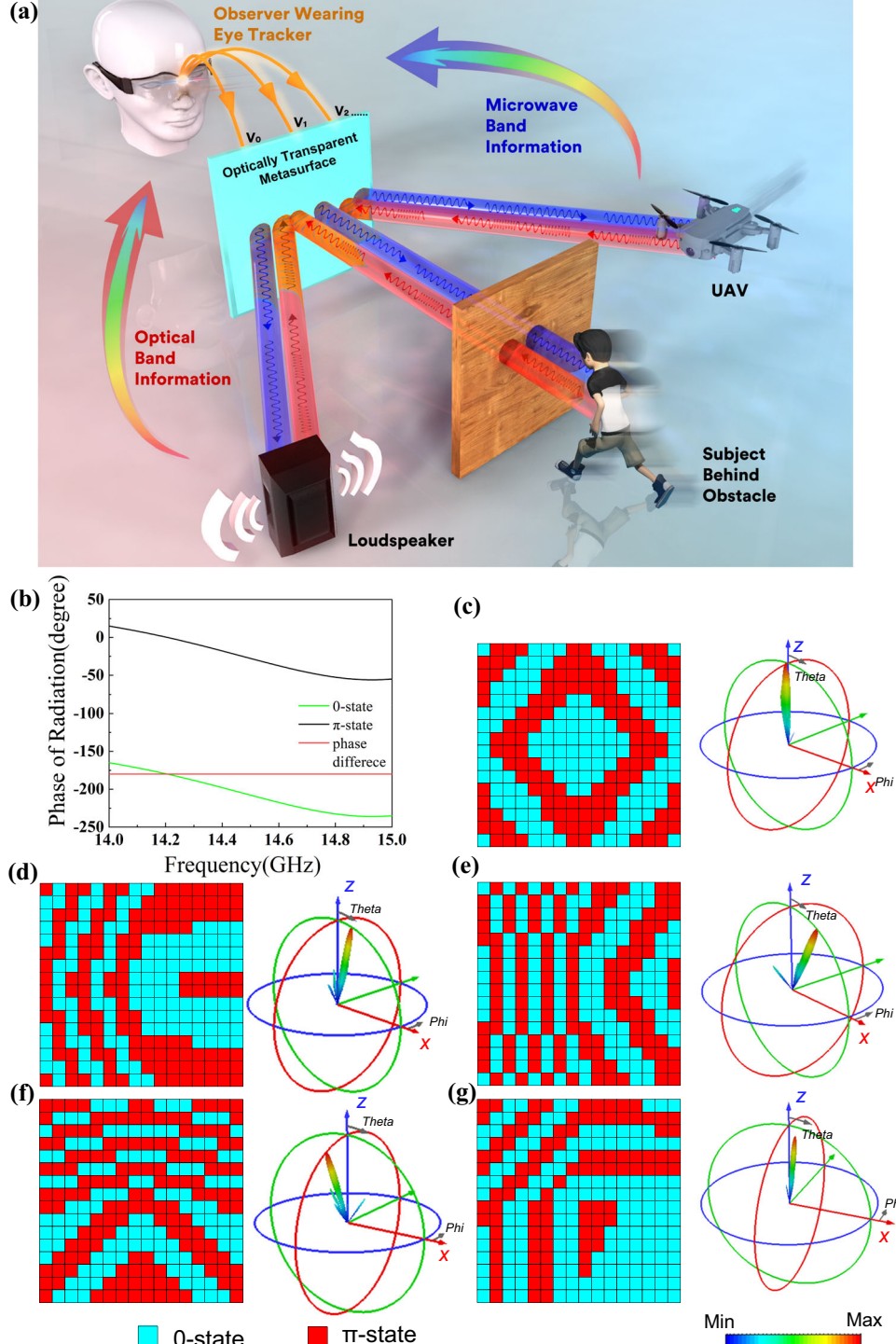

**Fig. 1 | Perception enhancement system based on vision-driven metasurfaces and its EM properties. a** Schematic of the perception enhancement system; **b** simulated radiation phase for 0-state and π-state; **c**–**g** simulated y-polarized far field patterns with an approximate scanning angle of (**c**)θ = 0°, φ = 0° (**d**) θ = 12.5°, φ = 0° (**e**) θ = 28°, φ = 0° (**f**) θ = 23°, φ = 270° (**g**) θ = 13.5°, φ = 315° with different coding sequences at 14.5 GHz.

optical transmittance. It enables metasurfaces to integrate information in multiple frequency bands and the integration of eyes (light) and metasurface (microwave).

On this basis, visual information in the optical frequency band is organically combined with the information in the microwave frequency band. Observers make basic decisions with the help of the observed optical information, and further mine more useful information in microwave frequency band. It makes humans more perceptive through receiving and processing of echoes and a rich set of functions have been developed. In this part, multi-targets' physiological state detection including respiration and heartbeat is carried out, followed by human location and motion detection under obstruction, target tracking based on visual information and health-care system for deaf people.

## Autonomous beam control following eye movements

In this scenario, we propose an eye-movement-based radiation beam steering scheme that the direction of irradiated EM waves changes with eye movement using eye tracker. In this method, autonomous and flexible beam control are realized which lays the foundation for microwave detection.

Here, we designed a radiation-type programmable metasurface. Two PIN diodes are integrated into each unit cell, the structure of which is illustrated in Supplementary Note S1. The designed metasurface consists of 16×16 unit cells with a total area of 240×240 mm². The equivalent RLC circuit of PIN diodes for simulation is illustrated in Supplementary Note S2. The metal pattern etched on top layer consists of integrated EM wave feed network, periodic unit cells and DC bias network. The DC bias network supplies DC voltage to diodes integrated on the metasurface units. The EM wave is stimulated at the center of the top layer and propagates through the integrated EM wave feed network which introduces an initial amplitude and phase distribution. To resolve the contradiction between optical transparency and EM loss, the metal pattern is constructed by metal meshes instead of metal patches. Such a design has little impact on the metal EM characteristics because the plasma-like property is kept by metal meshes[54].

To grasp a complete picture of the unit cell radiation spectra, the radiation phase of the unit cell under positive and negative voltages are simulated using CST Microwave Studio, as shown in Fig. 1b. Through reverse the biasing voltage of the PIN diodes, the phase difference of the two states (i.e. 0-state and π-state) maintains at around 180° with very small deviation within the operation band. In other words, the phase resolution of the proposed radiation-type unit is achieved through integrating two PIN diodes anti-symmetrically configurated. On this basis, each unit could implement the binary phase coded modulation based on the biasing voltage and thus constitute a 1-bit digital metasurface. In order to obtain specific radiation angle, the phase of each unit cell $\phi_{(x,y)}$ can be calculated according to the following equation,

$$\phi_{(x,y)} = k_0(x\,\sin\theta_0\,\cos\varphi_0 + y\,\sin\theta_0\,\sin\varphi_0) + \phi_0 \tag{1}$$

where $k_0$ is free space wave number, $(\theta_0, \varphi_0)$ is the designed radiation azimuth, $(x, y)$ is the unit cell coordinate, and $\phi_0$ is the initial phase introduced by EM wave feed network, $\phi_{(x, y)}$ is covert to a value between 0° to 360°. Since only binary phases are available for metasurface unit cells, we establish the mapping relationship between metasurface coding $CODE_{m,n}$ and $\phi_{(x, y)}$. Therefore, the coding sequence should be set as,

$$CODE_{m,n} = \begin{cases} 0, 0° \le \phi_{(x,y)} < 180° \\ \pi, 180° \le \phi_{(x,y)} < 360° \end{cases} \tag{2}$$

In this way, the radiation beam can be steered to the desired scanning angle. Since the bias network on the top layer of the metasurface affects the radiation of the metasurface, the radiation angle deviates slightly from the desired value. Figure 1(c)-(h) presents experiment far field radiation patterns with different coding sequences in representative scanning angles. It can be observed that good-shaped directional radiation beams are stimulated with very high pointing stability and metasurface has the ability to extend EM radiations to two-dimensional space. The proposed metasurface no doubt possesses the beam scanning capability for hemispherical coverage through simple reconfiguration of the code distribution. Due to the symmetry of the metasurface, radiation beams are also present in the symmetrical directions. In our demo, the desired scattering patterns and angles of the beam are designed to irradiate the AOIs.

Next, we combine the eye movement process with the metasurface to realize autonomous EM beam control. In the experiment, we investigated scenarios with eyes looking at targets in different positions over a period of time. Eye movement enables the control of radiation EM beam steering of the metasurface with the help of eye movement data processing program, FPGA and DAC module. The principle and method of eye tracking is illustrated in Supplementary Note S3. When the time of gazing on AOI exceeds the time threshold, the radiation beam will turn to the angle corresponding to the angle of sight. The threshold is set for subjects to continuously fixate the AOI for 1 second. Taking into account comfort zone of human eyes and metasurface radiation beam width, we set the metasurface EM beam horizontal angle range from −28° to 28°, and the maximum elevation angle to 23°.

The experiment environment and system are illustrated in Supplementary Note S4. The experiment results are depicted in Fig. 2, which illustrates the relation between gaze points as well as eye images in the upper half space and the measured far-field scattering pattern in the lower half space. Fig. 2a–e correspond to Fig. 1c to f, respectively. Using this control method, we successfully combine the field of view with microwave radiation of metasurface, making up for the limitation of visual perception and laying the foundation for microwave detection. It should be clarified that autonomous beam control system following eye movements have broad applicability to metasurface, not limited to specific metasurface design.

## "X-ray Glasses": Visible multi-targets' respiration and heartbeat detection

We have illustrated the autonomous beam control method following eye movements in the previous section. On this basis, a rich set of functions have been developed in this section. In order to improve humans' perceptual ability, an efficient and smart perception enhancement route named X-ray Glasses is proposed, associating the visual information with human physiological information through metasurface. The experiment is the detection of the respiration and heartbeat signals of multiple human targets in free space. The observer perceives not only visual images of the targets, but also their invisible physiological signals such as respiration and heartbeat.

Herein, two subjects were recruited to conduct the experiment, with the separation angle in azimuth direction set to be ±28°. Each of them wears a piezoelectric respiratory belt and disposable ECG electrodes connected to a portable electrocardiograph to record the reference respiration signal and ECG signal. During the measurement, volunteers were asked to remain stationary[55]. On the basis of autonomous beam control following eye movements, the observer sequentially gazes at one of the subjects over the time threshold in order to make EM waves irradiate the subjects. The detailed experimental configuration is illustrated in Supplementary Note S7(a).

When the person to be detected takes a breath, a portion of body parts, mainly the chest wall moves periodically according to his breathing pattern[56–58]. Moreover, the mechanical deformation caused by heart beating would also be transmitted to the chest wall, which is superimposed on the micro-movements caused by respiration. EM waves radiated onto the surface of human body can detect this faint micro-motion of the chest wall, which is recorded by the receiving antenna. The superposed fretting signal is then demodulated from the phase of echoes by utilizing the phase unwrapping signal processing method. After that, by applying a simple filtering finite impulse response (FIR) bandpass process, the respiration and heartbeat signals of the subject can be obtained. The theory of respiration and heartbeat detection is illustrated in Supplementary Note S5. Since the main lobe of EM wave radiated by metasurface is only facing one of the subjects each time, respiration and heartbeat signals of multiple targets are obtained in turn.

The experiment results of two subject's respiration and heartbeat detection are shown in Fig. 3. The red lines in the figures are time-domain waveforms of subjects' respiration and heartbeat signals

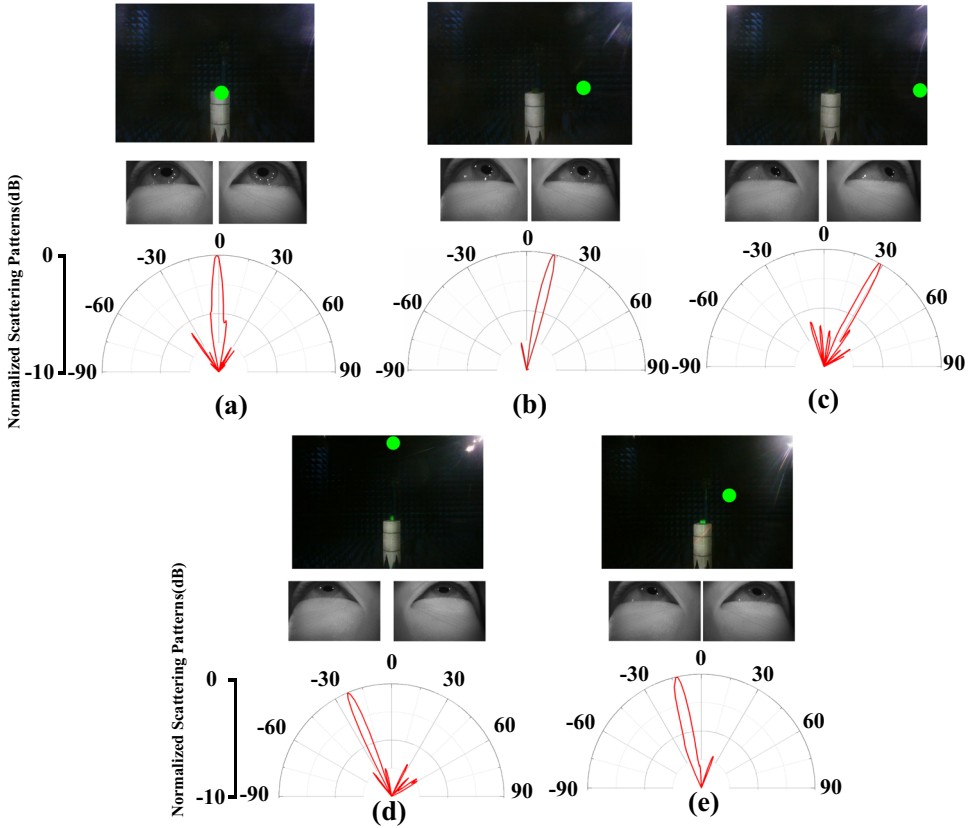

**Fig. 2 | Gaze point, eye images and measured metasurface far-field radiation spectra under different coding pattern. a–c** $\varphi = 0°$ **d** $\varphi = 90°$ **e** $\varphi = 135°$.

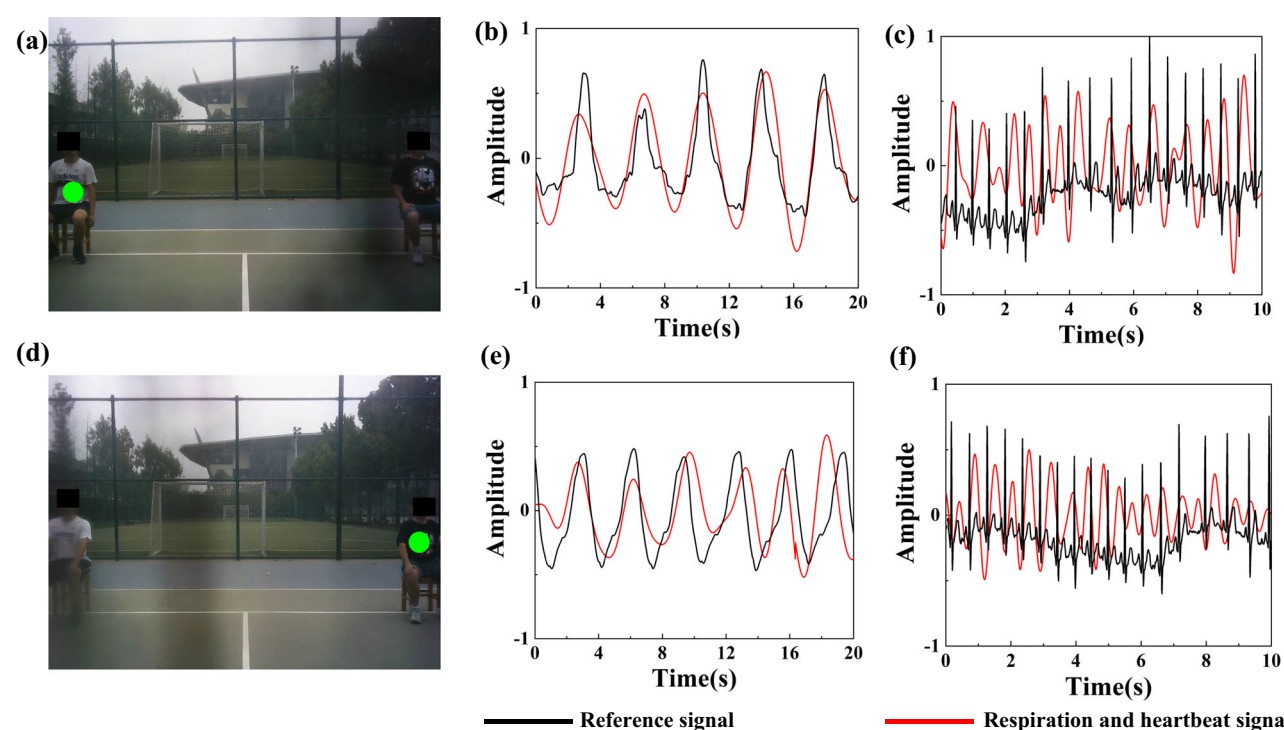

**Fig. 3 | Respiration and heartbeat signal detection of two subjects sitting at (−28°, 28°). a–c** Corresponds to the subject on the right. **d–f** corresponds to the subject on the left. **a**, **d** Experimental scenarios and the heatmap of gaze points. **b**, **e** Respiration signals of two subjects. **c**, **f** Heartbeat signals of two subjects.

collected using the vision-based metasurface platform, respectively. While the black lines are reference signals detected by the respiration belt and the portable electrocardiograph for comparison. Experimental results show that the respiration and heartbeat cycles matched well with the reference signals for both subjects. Moreover, the respiration and heartbeat signals of different subjects from different azimuth angles can be well separated on a physical level by means of changing the direction of EM waves through eye movement.

Therefore, it can be concluded that the system can effectively detect time-varying respiration and heartbeat signals of multiple human subjects. Compared with traditional methods, the design greatly reduces the system complexity. Moreover, it shows strong ability to suppress the environmental noises, clutters and multipath effects. Most importantly, the proposed method can well account for the traditional challenge of accurate breath and heartbeat detection when multi-target co-exists in the environment through beam regulation.

## "X-ray Glasses": Human location and motion detection behind plank obstacles

The experiment is the location and motion detection of human subjects behind obstacles. The proposed method can not only detect the exact location of hidden personnel in azimuth direction but also his movement pattern. Physiological signal detection, hidden location and motion detection provide a new complement to the perception of personnel besides visual information. In this section, plank obstacles were placed between observer and subjects to block the observer's view. One or three recruited volunteers were asked to stand in one of three azimuths (−28°, 0°, 28°) behind the plank. And the observer fixated the corresponding azimuth angle over the time threshold in sequence to determine whether the human subject exists in the specified location. Then the subject began to perform one of the following motions (jumping, squatting, falling, and walking) at the specified azimuth. The detailed experimental configuration is illustrated in Supplementary Note S7(b).

The localization of human subjects is achieved by detecting the respiration signal. When the EM beam controlled by the eye tracker is directed towards a specific azimuth, there exists a hidden human subject if the respiration signal can be detected. Otherwise, there presents no human subject at that azimuth. The theory and algorithm of human location as well as motion detection behind plank obstacles are illustrated in Supplementary Note S6.

The principle of motion detection can be described as follows: when the human subject conducts different types of motions, the body parts involved will produce characteristic micro-Doppler modulations to the incident EM waves. By applying short-time Fourier transform based time-frequency analysis to the collected scattered EM echoes, the time-varying instantaneous Doppler frequency of different body parts can be obtained. Since different motions modulate the incident EM waves in a different way, the obtained micro-Doppler signatures would show distinct distribution patterns in spectrograms. Thus, the resulted spectrograms can be utilized for distinguishing different motions. After obtaining the spectrogram, the principal component analysis (PCA) method is employed to extract the characteristic motion features. Then, the support vector machine (SVM) is utilized to classify different motions.

As depicted in Fig. 4a–c, apparent respiration signal is detected at azimuth of 28°, which indicates that a human subject exists in that direction. And it was found that the metasurface measured respiration signal is in highly accordance with that obtained using the contact respiration belt. While no breath is detected in other two azimuths, which is in line with the actual experimental setup. Fig. 4d–f show that respiration signals are detected in all three directions, which means there locates human subjects in all three directions. The above results validate the human location detection method.

Then the human target at an azimuth angle of 28° is chosen as an example to verify the motion detection performance. Fig. 4g–j show the obtained micro-Doppler signatures of the four behind-the-plank human motions. For in-place jumping motion, symmetrical micro-Doppler signatures appear in both positive and negative-hemi-axis, which caused by knees bending forward and hands waving backward when the human body jumps. The positive and negative Doppler frequency components of squatting are caused by the forward as well as up motion of one knee and backward as well as down motion of another knee, respectively. It is observed that the amplitude of positive and negative Doppler frequency components is much smaller than that of the jumping motion, but with increased repetition frequency. As for falling motion, characteristic volcano-shape positive Doppler signature envelope is sighted in the spectrogram, which is caused by the large scale forward falling motion of the body torso. Meanwhile, a small negative envelope appears in the spectrogram, induced by the backward kicking of foot after the falling. The walking motion presents obvious alternating positive and negative Doppler frequency, introduced by the EM wave modulation when subject walking towards and away from the metasurface. Then, based on these feature representations, a SVM classifier is trained to classify and recognize different behind-the-plank motions. Through training, the recognition accuracy of the model reached 94.6% for four behind-the-plank human motions.

Like breath detection, when detecting human motion of a specific azimuth, the narrow beam emitted by metasurface can exclude the interference from other positions in the detection space, which made it capable of robust operating in a real detection scenario where multiple human targets in motion co-exists. By this means, human location and motion can be detected without being visual sighted by the observers, that is, the observer has a pair of X-ray Glasses.

## "Glimpse-and-forget" metasurface smart target tracking system

When maintaining high-intensity work in complex scenes or performing multiple tasks, the visual system can only selectively handle a small part of visual information from a large number of candidates. It restricts the visual system's capability to attend and process the large amount of information simultaneously present within a visual scene. In this situation, visual system may focus on one goal while ignoring others, leading to the miss of useful information. Aiming at this issue of human visual system, a beam tracking method based on vision is proposed in this scenario to improve the efficiency of information acquisition and processing. Once a target is selected by human vision system, the EM beam can automatically track the target without further attention, while visual system is freed up to process other information instead of continuing to track the selected target. By this means, visual perception is enhanced with the assistance of eye tracker. And the burden on the vision system is reduced, allowing more tasks to be performed at the same time.

To track the object over time, we follow the scheme of the Kanade-Lucas-Tomasi (KLT) algorithm to create a tracker that automatically tracks a single object. The KLT tracker is commonly used for tracking feature points of a target due to its excellent processing speed and high accuracy. It is a classical algorithm widely applied in applications such as video stabilization and image mosaicing. By means of invoking the eye tracker's camera, the KLT algorithm can track a set of feature points across consecutive video frames. The tracking procedure is denoted as follows: First, the observer locates the object to be tracked through blinking eyes quickly. The eye movement data processing program applies a bounding box to the object. Next, the feature points can be reliably extracted for the target object within the box. KLT tracker confirms the motion of the target object by tracking these feature points in frames. Finally, the eye tracker to metasurface interface program determines the reaction of the metasurface by itself. To be specific, digital-coding sequences of the programmable

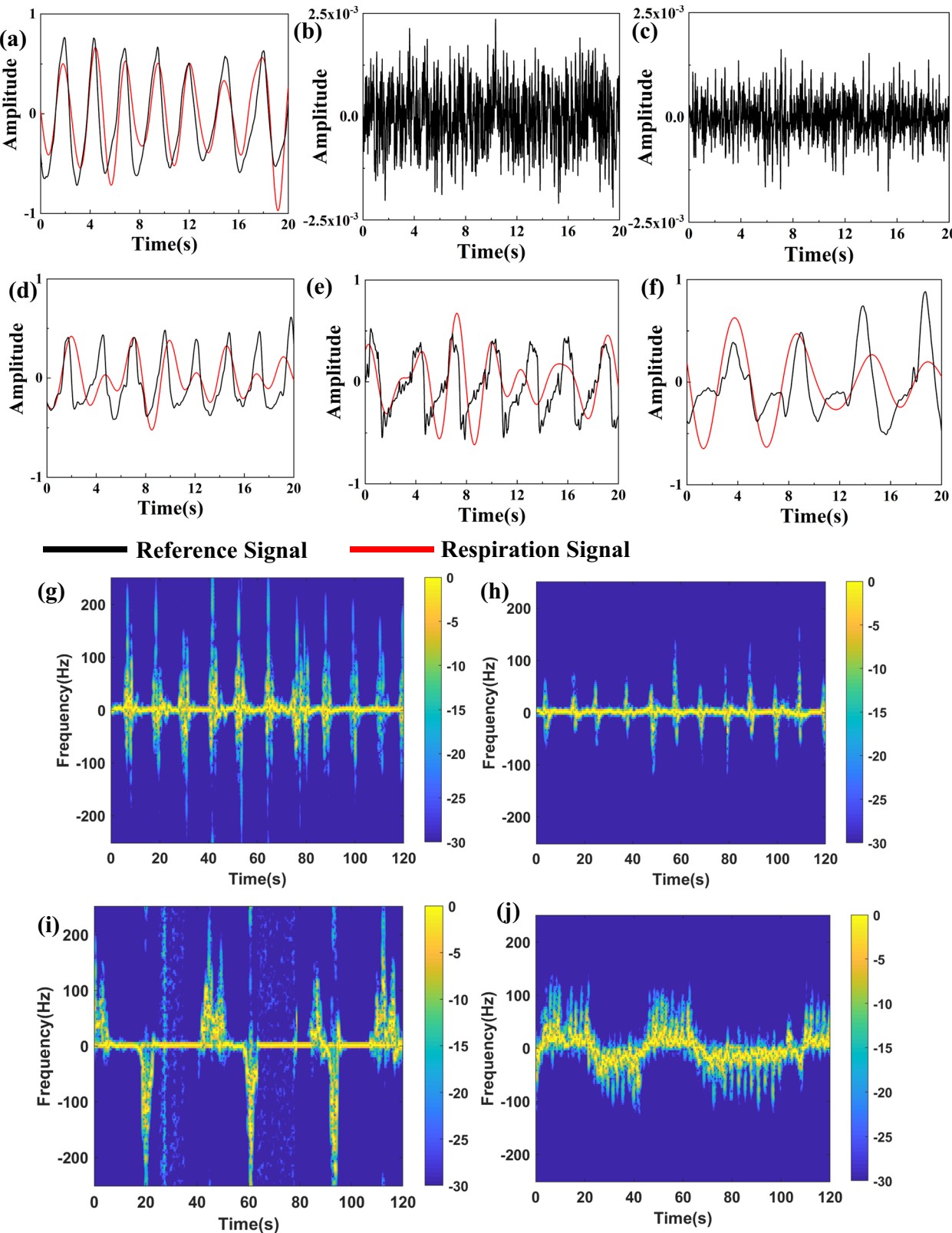

**Fig. 4 | Location and motion detection of human subjects behind plank obstacles.** Respiration detection results for the case where only one subject exists when observer fixate the azimuth of (**a**) 28°, (**b**) 0°, (**c**) −28°, respectively. Respiration detection results for the case where three subjects co-exists when observer fixate the azimuth of (**d**) 28°, (**e**) 0°, (**f**) −28°, respectively. Spectrograms of four behind-the-plank human motions: (**g**) jumping (**h**) squatting (**i**) falling (**j**) walking.

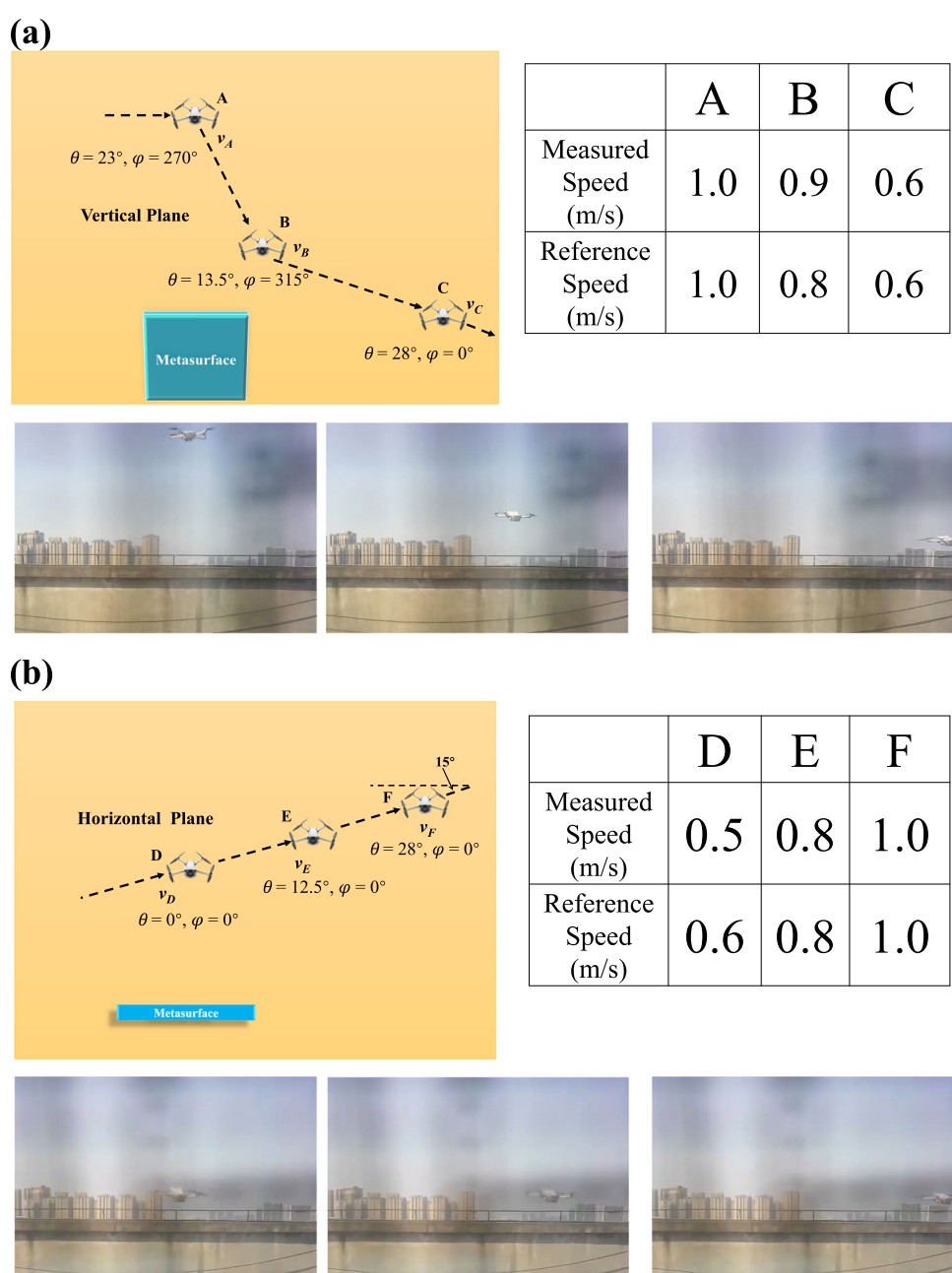

**Fig. 5 | The flight trajectory, and reference speed when the UAV flies. a** Vertical plane. **b** Horizontal plane.

metasurface are changed to keep the radiation beam turning to the angle corresponding to the tracked object. In order to further reduce the switching time, we pre-select a series of EM wave radiation angles according to the far-field experiment results. When the target approaches the selected radiation angle, the EM wave will illuminate the corresponding angle in advance. The radiation angle can be tuned depending on the application scenario.

We consider a series of applicational situations in multi-targets or complex environments, e.g. unmanned aerial vehicle (UAV) and departure hall. In the departure hall, crowds of people walk with luggage at railway stations or airports, which brings difficulty to tracking. Using smart metasurface platform, observers can lock onto targets in large crowds without affecting the search for the next target, and microwave beams can then track them based on the information provided by KLT tracker, recording the target's locations and assisting in detecting target states. As another example, we need to track

slow-flying UAVs and jam them with the help of microwave in the outdoor environment. Through blinking eyes quickly, the programs can select, lock and track the gaze area where the UAV appears. Then the microwave beam is steered to irradiate and track the UAV. Two intentional vulnerabilities of UAVs, Jamming and navigation spoofing, can be exploited in this way. EM waves incident to UAVs can carry high-power noise or counterfeit navigation signals in order to hinder a navigation service or produce fake positions. Therefore, it is possible to successfully spoof or jam a civilian UAV and change its flight trajectory from the predefined ones without the user's notice.

Herein, we take the UAV scene as an example. In order to prove that EM waves can illuminate the tracked target, the most intuitive way is to obtain real-time information from the tracked target. In the present embodiment, the velocity of the UAV is collected for verification. The experiment scenario and principle of velocity measurement are illustrated in Supplementary Notes S8 and S9, respectively. Figure 5

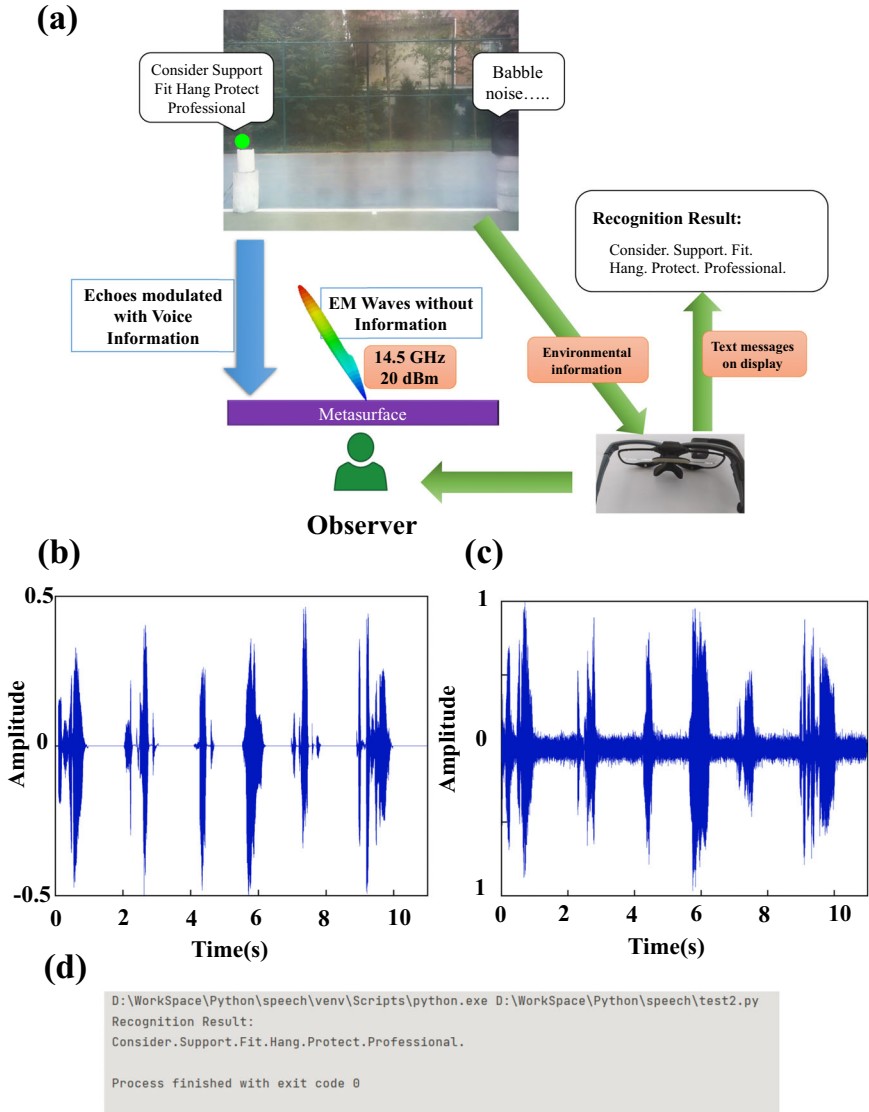

**Fig. 6 | Barrier-free speech acquisition and enhancement system. a** Schematic of barrier-free speech acquisition and enhancement system. **b** Waveform of raw recorded speech. **c** Waveform of enhanced microwave speech signal. **d** Speech recognition result using speech recognition module.

illustrates the flight trajectory of UAV and compares the measured speed as well as reference speed, which verifies the effectiveness of the design. It should be noted that this example is to illustrate the design concept and implementation method. The purpose of the glimpse-and-forget metasurface smart system is not only to measure the velocity of the UAV, but to prove that EM waves can irradiate and then track it.

**Barrier-free speech acquisition and enhancement system for deaf people**

Speech is an essential tool in information sharing between people. Deaf people have serious problems to access information due to their inherent difficulties to deal with spoken languages, contributing to a reduction of social connectivity and is associated with increased morbidity and mortality. Therefore, it is of great significance to improve the speech perception ability for deaf people.

In this scenario, a barrier-free speech acquisition and enhancement system is proposed to provide healthcare access and overcome communication barriers for deaf people based on vision-driven metasurface platform, as shown in Fig. 6a. A deaf person observes an interested target driven by the voice vibrates (loudspeaker in this case). The radiation signal of the target is measured and analyzed,

thereby obtaining voice signals and the corresponding speech recognition results. With the help of the wearable display attached to eye tracker, deaf people can see text messages projected on his eyes corresponding to voice signals. In this way, sound wave information is converted into microwave information, and then into visual signal directly visible to the human eyes. The system breaks through the barriers of visual information and auditory information, realizing the visibility of speech signals. It helps deaf people to acquire speech information without barriers, so as to communicate, watch dramas and movies, especially in noisy environments.

Traditional acoustic microphones and recognition algorithms are easily to be affected by environmental noise and multiple sound sources. The system, however, is capable of excluding the babble noise from other positions and empower the deaf people with cocktail party effect, that is, the ability of human hearing sense to extract a specific target sound source from a mixture of multiple sound sources and background noises in complex acoustic scenarios. Moreover, different from other reported metasurface based speech acquisition work, the system can obtain the understandable clear speech signal by analyzing the reflected EM wave, instead of a series of simple instructions.

Experiments were conducted to verify the effectiveness of our speech acquisition and enhancement system. The detailed experimental configuration is illustrated in Supplementary Note S10. Metasurface platform first directs EM beam guided by eye tracker toward the loudspeaker to detect its surface vibrations driven by voice signal. The backscatter EM signals is collected using a receiving antenna connected to vector network analyzer (VNA). The micro-displacement information of audio signal is carried in the phase of the backscattered EM signal.

To reveal the authentic audio signal, the minimum mean-square error (MMSE) based short-time spectral amplitude (STSA) speech enhancement algorithm is applied to suppress the noise and enhance speech information. The theory and algorithm of speech acquisition and enhancement are illustrated in Supplementary Note S11. Fig. 6b denotes the waveform of a recorded audio played by the loudspeaker, an adult female saying: Consider, Support, Fit, Hang, Protect, Professional. Raw microwave speech echo signals disturbed by the ambient noise are recorded by VNA. On this basis, MMSE-STSA algorithm is applied to obtain clean voice information, and then the enhanced audio signal is further processed using a python-based speech recognition module. The speech enhancement result and the recognition result are illustrated in Fig. 6c and d. The recognition perfectly reconstructs the text information from the target audio source, which proves the effectiveness of our proposed metasurface based speech acquisition and enhancement system. We have attached the original voice signal (Supplementary Audio 1) and the voice signal after noise removal measured by metasurface (Supplementary Audio 2).

## Discussion

In this work, we propose the concept of vision-driven metasurface and on this basis present a framework for perception enhancement. A new control manner from the subject's consciousness to the metasurface pattern is established by means of combining the eye tracker and the programmable metasurface. To illustrate our idea, we presented a specific intelligent metasurface to adjust the radiation beams. Embedded with human visual perception, the proposed intelligent metasurface platform exhibits effectiveness in response to an ever-changing AOI and the state of the subject. In this foundation, typical schemes with distinct functions are demonstrated including a physiological-index-monitoring system, an X-ray-glasses system, a glimpse-and-forget tracking system and a healthcare system for the deaf people.

Compared with traditional methods, signal processing, computer vision, EM and other technical means are fused in the platform. The introduction of metasurfaces in the intelligent platform enables humans with stronger information acquisition, processing, and perception ability. Perception of multi-band information, such as optical and microwave bands, compensates for the shortcomings of human vision under obstruction. Moreover, this method can help humans identify target of interest and assist target-tracking. Furthermore, by utilizing human visual perception and judgment ability, the proposed method allows the metasurface and its intelligent platform with more degrees of freedom as well as more advanced functions, which offers a new definition of a metasurface and paves the way towards cognitive and intelligent metasurfaces.

## Methods

### Hardware, simulation and sample fabrication

The simulation of the unit cell and metasurface in time-domain solver was conducted using commercial software CST Microwave Studio 2019. MADP-000907-14020 commercial PIN diodes were used in the metasurface unit cell. The configuration of computer is Intel(R) Core(TM) i7-9750H CPU @ 2.60 GHz /32GB/2 T SSD. The eye tracker is conducted using ASee glasses eye movement system from 7invensun.

### Experimental measurement

All experimental procedures involved were performed according to protocols approved by the Fourth Military Medical University. An informed consent was given by all participants. Each participant was given 300 yuan in the experiment. The experiment aims to validate the perception enhancement capabilities of visually-driven metasurfaces and therefore does not involve the gender or sex analysis.

The experimental setup is shown in Supplementary Notes S4, S7, S8 and S10. All experiments require an observer to gaze at the specified azimuth angle over a period of time. The experimental configuration for autonomous beam control via eye movements is described in Supplementary Note S4. The measurement system consisted of eye track measurement and EM wave measurement. The experimental process was performed in an anechoic chamber.

The experimental implementation details for X-ray Glasses are described in Supplementary Note S7. The experiment includes two major parts: visible multi-targets' respiration & human location and motion detection behind plank obstacles. Subjects in visible multi-targets' respiration experiment section included 2 males (ages 25, 182 cm, 76 kg and ages 33, 172 cm, 73 kg). Subjects in human location and motion detection behind plank obstacles section included 20 males and 5 females. The subjects ranged in age from 20 to 41 years old, ranging in height from 1.62 m to 1.92 m, weights ranging from 44 kg to 89 kg.

The experimental configuration for glimpse-and-forget metasurface smart target tracking system is described in Supplementary Note S8. The experimental implementation details for speech acquisition and enhancement system are described in Supplementary Note S10.

### Reporting summary

Further information on research design is available in the Nature Portfolio Reporting Summary linked to this article.

## Data availability

The authors declare that all relevant data are generated in the study are provided in the Supplementary Information/Source Data file.

## Code availability

The code used for data processing is available from the corresponding author on request.

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

## Acknowledgements

T.Q. and Q.A. acknowledges the financial support from the National Natural Science Foundation of China (Grant Nos. 62101589 and 62301568). T.Q. acknowledges the financial support from the Open Project of State Key Laboratory of Millimeter Waves (Grant No. K202404).

## Author contributions

T.Q. and Q.A. conceived the idea and performed the numerical simulations. T.Q., Q.A., L.C., and S.L. performed the experiment. M.C. and H.L. wrote the manuscript. S.Q. and J.F.W. drew the graphs in the manuscript. T.Q., Q.A., J.Y.W. and C.Q. discussed the results and commented on the manuscript. J.Q.W. and C.Q. supervised the project.

## Competing interests

The authors declare no competing interests.
