## [Peer Review File · Nature Communications]

Vision-Driven Metasurfaces for Perception EnhancementREVIEWER COMMENTS

Reviewer #1 (Remarks to the Author):

In the manuscript "Vision-Driven Metasurfaces for Perception Enhancement", by T. Qiu et al., the authors propose and demonstrate a metasurface-based platform for visual perception enhancement by monitoring eye movements using conventional eye tracking and then controlling metasurfaces that operate in the microwave regime. Various demonstrations are shown with the platform. The basis of the work is a combination of a programmable beam steering metasurface in the microwave region, and a number of conventional technologies and algorithms, such as the eye tracking system to choose the beam steering angle and speech recognition manipulation and software. The work is somewhat similar to previous work from the group (eLight 2, 10 (2022)), albeit with a different control method. Overall, I think the work is interesting and noteworthy enough to be published in Nature Communications, however it needs major revisions and improvements. In particular the following comments should be addressed:

1. Regarding the eye tracking and metasurface integrated system, it seems that the eye tracking is working only in the horizontal plane. Is this a limitation of the system? Could the authors comment about extending the system to full space including up and down as well as left and right?
2. For breathing and heartbeat measurements, what is the limitation for the metasurface? Both subjects seem to show an extremely healthy 60 bpm for their heartrate, but the measured sin curves are quite wide, so could result in some overlap if the responses are closer in time. Is there an upper limit for the measurements of breathing and heartrate?
3. The heartbeat is known to be a 'lub-dub' sound, so what exactly is the metasurface measuring for each peak and valley in the signal? Is it a single sine curve for each complete beat, so the microwave response of this lub-dub is a single peak? Could the authors give some detail or comments?
4. For the respiration and heartbeat detection, could the authors comment on and provide any experiments or calculations about how far away the subject can be? Does the thickness of their clothing matter? How about their orientation? Can the measurements be performed from the side or back of a subject with the same accuracy?
5. For the tracking, a video was watched on a screen. Is that a fair experiment? In the real-world, the drone would be moving in 3D, away from the tracking user, as well as in the horizontal plane.
6. In terms of the speed of tracking, the authors should discuss the speed of modulation of the metasurface. If it is much slower than the movement of the target, the target would already be away from the position by the time that the beam is steered to that angle.
7. Overall, I am not sure what the tracking demonstration is showing that is more than the simple examples of beam steering control using the eye tracker in Figure 3.
8. For speech recognition from microwaves, similar to an earlier comment, is the information still available if the speaker was facing the opposite way? We would be able to hear it with our ears, but does the metasurface need direct line of sight to detect the surface vibrations? If so, for a human's voice, would the metasurface have to be focused directly on the mouth?
9. It would be interesting to hear the original voice and the recovered one from the reflected microwaves to compare them, as the waveform looks like it contains a lot of noise.
10. The authors should discuss the practicality of the platform that uses microwave frequencies with metasurfaces. The 40x50cm metasurface would be fairly large for a user to carry around. Additionally, the user would need an antenna along with other equipment in order to detect the reflected microwaves. Could this be integrated into a single metasurface that emits and receives microwave signals?
11. Finally, the manuscript could do with an overall grammar check to improve the English presentation. The first word of the introduction needs an article to start with, and there are numerous other small mistakes that need to be cleaned up. Regarding Figure 4, the authors write 'the black lines', but they are blue, and there are various spelling mistakes such as 'matasurface' in the SI. It also seems that the radiation spectrum of Figure 3a is incorrect, it is going in the same direction as (b). Figures 5e and f also appear to be the same data.

Reviewer 2 (Remarks to the Author):

Authors attempt to explore so-called vision-driven metasurface for the purpose of perception enhancement. In this work, the eye movement is detected with a vision eye-tracking glass and then is used to control a dynamic metasurface for sensing the target. I notice that the similar idea has been explored in following two publications, i.e.,

<https://elight.springeropen.com/articles/10.1186/s43593-022-00019-x>

<https://elight.springeropen.com/articles/10.1186/s43593-022-00016-0>

In the above two publications, the brain signal is recognized using a specialized machine, and then to manipulate the metasurface for the purpose of sensing or communication. Although different signals (eye-movement sign, or brainwave signs) are considered, the underlying operational mechanism and procedures are almost the same for them. In addition, although a collection of experiments is presented, the trivial integration of vision and metasurface is just considered and the in-depth principle and mechanism are lacking for the integration of eye (vision) and metasurface (microwave). For these reasons, I would not like to recommend its publication in Nat. Comms.

Other comments are following.

- 1) The details of eye tracking glasses used in this work are required for readers to reproduce the reported results.
- 2) Definitely, it needs to add more details (like, the time, power, accuracy, and so on) for the whole procedure (like Fig.7a), i.e., the recognition of eye movement, the control of metasurface, the perception of target, and the visualization of the target on the glasses.
- 3) As for the section of vision multi-targets' respiration and heartbeat detection, what is the maximal number of targets the proposed system can achieve? And, is there any constraints on the distance between the targets? In the presented results, only static targets are considered.

How to deal with the moving targets?

- 4) As for the “X-ray eyes”, it is not suitable since the eye has no ability of seeing through the obstacle. Maybe, the X-ray glasses is more suitable.
- 5) As for the section of barrier-free speech, I suggest to remove this part, since the metasurface here plays the communication relay, as intensively discussed in the area of wireless communication, i.e., RIS.
- 6) The figures need to be improved, because many legend and marker are blurred and not clear to me.
- 7) It'd better if Fig. 1 and 2 are combined together.
- 8) Why is the frequency (4.0GHz) chosen?

Reviewer #3 (Remarks to the Author):

The authors proposed microwave beam steering metasurfaces with eye tracking functions. By controlling digital-coding sequences, the device modulates beam propagation. Based on this approach, the authors demonstrate respiration and heartbeat signal detection, subject detection located behind obstacle, and UAV tracking and so on. The proposed vision-driven metasurface technically sounds, but I could not see substantial improvement in metasurface design and its operating systems (concept of coding metasurfaces) compared to many other prior arts. In this regard, I cannot recommend this paper for the consideration in the Nature Communications.

Reviewer #1 (Remarks to the Author):

In the manuscript “Vision-Driven Metasurfaces for Perception Enhancement”, by T. Qiu et al., the authors propose and demonstrate a metasurface-based platform for visual perception enhancement by monitoring eye movements using conventional eye tracking and then controlling metasurfaces that operate in the microwave regime. Various demonstrations are shown with the platform. The basis of the work is a combination of a programmable beam steering metasurface in the microwave region, and a number of conventional technologies and algorithms, such as the eye tracking system to choose the beam steering angle and speech recognition manipulation and software. The work is somewhat similar to previous work from the group (eLight 2, 10 (2022)), albeit with a different control method. Overall, I think the work is interesting and noteworthy enough to be published in Nature Communications, however it needs major revisions and improvements. In particular the following comments should be addressed:

1. Regarding the eye tracking and metasurface integrated system, it seems that the eye tracking is working only in the horizontal plane. Is this a limitation of the system? Could the authors comment about extending the system to full space including up and down as well as left and right?

Author response:

Thank you very much for the instructive comment and advice. In order to enrich the application scenarios, improve system integration as well as extend the fullspace beam scanning capability, the entire metasurface structure has been modified by a big

margin. Fig. R1 proposes the schematic and architecture of the newly designed metasurface structure.

Figure R1. Overall configuration of the proposed radiation-type metasurface with 16×16 units. (a) perspective view and (b) lateral view. (c) Metasurface unit cell topology, the dimensions of the unit cell are $l = 15$ mm, $l_1 = 3.6$ mm, $l_2 = 4.4$ mm, $l_3 = 0.8$ mm, $l_4 = 1.1$ mm, $l_5 = 1.8$ mm, $l_6 = 0.3$ mm, $l_7 = 0.4$ mm and $l_8 = 0.1$ mm.

As optically transparent substrates, the dielectric materials polyethylene terephthalate (PET) and polycarbonate (PC) are adopted as substrates. The top and bottom layers are filled by PET substrate with a height $h = 0.125$ mm, dielectric constant $\epsilon_r = 3.4$ and tangent loss $\tan \delta = 0.001$, respectively. The middle layer is a PC layer with a thickness of 0.8 mm, which is used to support and isolate the metallic patterns. PC and PET are bonded by UV-curable adhesives to meet the requirements of light transmittance and processing temperature. The metasurface is designed to operate in 14.5 GHz with a size of 240×240 mm².

To resolve the contradiction between optical transparency and EM loss, the metal pattern is constructed by metal meshes instead of metal patches. Such a design has little

impact on the metal EM characteristics because the plasma-like property is kept by metal meshes. The metal meshes etched on the bottom layer act as the metal ground.

The metal pattern etched on the top layer consists of an integrated EM wave feed network, periodic unit cells and a DC bias network. There are 16×16 periodic cells on the top layer, each consisting of two rectangular copper meshes that are bridged to the EM wave feed network by two PIN diodes. The PIN diodes anti-symmetrically integrated in unit cells are MADP-000907-14020 from MACOM Technology. The DC bias network supplies DC voltage to the diodes integrated on the metasurface units. The entire DC feed circuit loop is set up by sharing part of the network with the integrated EM wave feed network. A total of 512 PIN diodes are controlled individually by the steering-logic board via 256 independent narrow-width (0.1 mm) bias lines. Chip inductors of 82 nH are used in the DC feed circuit loop in order to minimize the impact of DC bias network on the metasurface's radiation performance and ensure a flat gain response. The EM wave is stimulated at the center of the top layer and propagates through the integrated EM wave feed network which introduces an initial amplitude and phase distribution. Impedance matching is used at the junction between the feed network and unit cell to improve operation bandwidth and loss efficiency.

The phase resolution of the proposed radiation-type unit is achieved through integrating two anti-symmetrically configured PIN diodes. The radiation phases of the unit cell under positive and negative voltages are simulated using CST Microwave Studio, as shown in Fig. R2(a). It is clear that through reverse the biasing voltage of the PIN diodes, the phase difference of the two states maintains at around 180° with very

small deviation within the operation band. On this basis, each unit could implement the binary phase coded modulation based on the feeding power. The radiation angle can be tuned through controlling the coding sequence via the steering-logic board.

Figure R2. (a) Simulated radiation phase for "0-state" and " π -state". (b-f) Simulated y-polarized far field patterns with an approximate scanning angle of (b) $\theta = 0^\circ$, $\varphi = 0^\circ$ (c) $\theta = 12.5^\circ$, $\varphi = 0^\circ$ (d) $\theta = 28^\circ$, $\varphi = 0^\circ$ (e) $\theta = 23^\circ$, $\varphi = 270^\circ$ (f) $\theta = 13.5^\circ$, $\varphi = 315^\circ$ with different coding sequences at 14.5 GHz.

Fig. R2 (b)-(f) presents simulated and experiment far field radiation patterns with different coding sequences in representative scanning angles. It can be observed that good-shaped directional radiation beams are stimulated in the designed directions with very high pointing stability. The proposed metasurface no doubt possesses the beam scanning capability for hemispherical coverage through simple reconfiguration of the code distribution. Moreover, due to the symmetry of the metasurface, radiation beams

are also present in the symmetrical directions.

It can be seen that the entire metasurface structure has been greatly modified in revision, which enriches the application scenarios and improves the system integration. The improvements compared to the original manuscript are as follows: First, the original architecture can only realize EM beam scanning in one-dimensional space. While new metasurface architecture can extend EM radiations to two-dimensional space. The proposed metasurface no doubt possesses the capability of beam scanning for hemispherical coverage by simply reconfiguring the code distribution in the metasurface. Secondly, the metasurface's operating mode is changed from reflection-type to radiation-type, which greatly simplifies the metasurface structure. The transmitting antenna is removed, that is, allowing the metasurface to perform all transmitting tasks. Thirdly, by using transparent dielectric substrates and metal mesh patterns, the proposed metasurface structure can achieve high visible optical transmittance. It enables metasurfaces to integrate information in multiple frequency bands and the integration of eyes (light) and metasurface (microwave). Observers make basic decisions with the help of the observed optical information, and further extract more useful information in the microwave frequency band. In addition, by means of redesigning, the metal patterns are only etched on top and bottom layers, and the structure gets rid of complex through-holes and vias. The size of the metasurface is reduced from $50 \times 50 \text{ cm}^2$ to $24 \times 24 \text{ cm}^2$. The reduced size of transparent metasurface can be compared to that of human face, which creates the prerequisites for transparent lenses or masks. Over all, the proposed metasurface architecture has better EM

performance, smaller size, simpler structure, higher integration and richer application scenarios, which lays a solid foundation for perception enhancement system.

2. For breathing and heartbeat measurements, what is the limitation for the metasurface?

Both subjects seem to show an extremely healthy 60 bpm for their heartrate, but the measured sin curves are quite wide, so could result in some overlap if the responses are closer in time. Is there an upper limit for the measurements of breathing and heartrate?

Author response:

When measuring respiration and heartbeat with metasurface, there are two main limitations of the metasurface. First, the operating frequency of the metasurface. Second, the beam width of the radiation pattern. The higher the working frequency of the metasurface, the more accurate the measured body surface micro-movement signal is. The larger the metasurface array aperture is, the narrower the corresponding beam is, and the less the clutter and interference of respiration and heartbeat signal is measured. However, in practice, considering the manufacturing costs and the requirements of antenna low profile, we usually do not increase the frequency or increase the aperture size without limitation. The redesigned metasurface in this work has a working frequency of 14.5 GHz and an aperture size of 24*24 cm², which reach a good balance.

The metasurface system measures the mechanical deformation on the body surface produced by heartbeat and breathing, then extracts the time-varying respiration and heart sinusoidal waveform from the mechanical deformation. The width of the

sinusoidal waveform is different for different individuals with different respiratory rate and heart rate. Individuals with higher breathing and heart rate have narrower width of their sinusoidal waveforms. Similarly, if an individual's heart rate increases, the sinusoidal curve narrows accordingly. However, the respiratory rate and heart rate of the human target have their typical frequency range. Therefore, as long as the sampling rate is high enough, the signals of breathing and heartbeat micro-movements on the body surface can be completely sampled. The sampling rate is set according to the highest human heart rate, and the sampling rate of the system is reasonably designed, so that the collected echoes will not be overlapping.

As the reviewer points out, there is an upper limit to the use of metasurface measurements of respiration and heartbeat, so the sampling rate of the metasurface must meet the Nyquist sampling theorem, that is, the sampling rate must be twice the upper limit of the heart rate, in order to truly sample the respiration and heartbeat waveform. In this work, the upper limit of sampling is 91 Hz, which is much larger than the heartbeat and respiration, so it can meet the normal requirements of respiratory heartbeat test.

As a verification, we conducted the following experiment. In this experiment, the target first performed distance exercise, ran, and then entered the scene to detect breathing signals. With the passage of detection, the target's breathing gradually became calm. We recorded the respiratory and heartbeat signal of the whole process, as shown in the Fig. R3. It proves that the metasurface system is suitable for high respiratory (>50 per minute) and heart rate (>100 per minute) measurement.

Figure R3. (a) Respiration and (b) heartbeat signals of the subject after exercise.

3. The heartbeat is known to be a ‘lub-dub’ sound, so what exactly is the metasurface measuring for each peak and valley in the signal? Is it a single sine curve for each complete beat, so the microwave response of this lub-dub is a single peak? Could the authors give some detail or comments?

Author response:

The authors thank the reviewer for his valuable comment. Due to the exist of heart beating, mechanical deformation, i.e., the displacement of the heart deformation is produced on the body surface, which is measured by metasurface system. The measured peak surface of heartbeat signal corresponds to the maximum mechanical deformation (outward) at the end of the diastolic period, while the trough corresponds to the mechanical deformation (inward) at the end of the systolic period. Therefore, a complete period of the heartbeat signal measured by metasurface is approximately the period of a sine wave, which is the period of the deformation of the body surface corresponding to a cardiac cycle. In other words, the period from the start of one heartbeat to the start of the next. The heartbeat signal measured by metasurface reflects

the cardiopulmonary function of the human target. A normal person's heart rate is 60-100 beats per minute when he is calm. More details related to heartbeat signal are added in revision.

Fig. R4. a typical example of the heart sound signal waveform measured by the contact sound sensor.

When blood is pumped through the heart's chambers, it makes a distinct noise that's often described as heart sound. The heart sound refers to the sound produced by the vibration caused by the contraction of the myocardium, the closing of the heart valve and the impact of blood on the wall of the ventricle and the wall of the major artery. The intensity, frequency and relationship of heart sound can reflect the function of heart valve, myocardium and blood flow in heart. Each cardiac cycle can produce four heart sounds, and the first and second heart sounds are generally heard, i.e., "lub-dub". The first heart sound occurs during systole, marking the beginning of ventricular systole. The second heart sound occurs during the diastole phase and marks the beginning of the ventricular diastole phase. Fig.R4 shows a typical example of the heart sound signal waveform measured by the contact sound sensor. It can be obviously observed that the heart sound signal does not have the characteristics of sine wave, and

its frequency is higher than that of the heartbeat signal. Usually, researchers do not measure the displacement of heart sounds on the body surface, but use contact sound sensors to directly measure the intensity and frequency of heart sound signals.

4. For the respiration and heartbeat detection, could the authors comment on and provide any experiments or calculations about how far away the subject can be? Does the thickness of their clothing matter? How about their orientation? Can the measurements be performed from the side or back of a subject with the same accuracy?

Author response:

For a typical radar system, the farthest detection range R_{max} is determined by the radar equation, as shown in the following formula,

$$R_{max} = \left[\frac{P_t G_t G_r \lambda^2 \sigma}{(4\pi)^3 S_{imin}} \right]^{1/4} \quad (R1)$$

where P_t is the transmitting power with a value of 0 dBm, G_t and G_r are transmitting and receiving antenna gain an approximate value of 10 dBi, respectively. λ is the operation wavelength with a value of 2.07 cm. σ is the RCS of the target, an RCS of 0.3 m² is as a typical value for a human chest. S_{imin} is the minimum detectable signal, that is, the receiver sensitivity.

$$S_{imin} = K T_0 B F_n (SNR)_{omin} \quad (R2)$$

where K is the Boltzmann constant with a value of 1.38*10⁻²³ J/K, T_0 is the reference temperature with a value of 290 K, B is the receiver bandwidth with a value of 500 Hz, F_n is the receiver noise factor, and $(SNR)_{omin}$ is the required signal-to-noise ratio. As a rule of thumb, we suppose $F_n = 15$ and $(SNR)_{omin} = 20$ dB. By

converting the above values into the International System of Units and substituting the above values into Equations R1 and R2, the theoretical farthest detection range $R_{max} \approx 37$ m can be obtained. In the actual experiment, many factors including transmitting power, RCS of the target, temperature can have great influence on the farthest detection range R_{max} . The farthest detection range is the result of the combined action of the above factors. Typically, the detection range is much smaller than the maximum value because real experimental conditions are not ideally perfect. If a longer detection distance is required, the most common and easiest method is to use a low-noise amplifier to increase transmitting power P_t .

The thickness of the clothes will not affect the detection results, because the dielectric constant of the fabric is relatively low, and it is a non-polar material. The EM wave signal can directly penetrate the clothing and irradiate the chest wall of the human body, capturing the slight movement of the chest wall caused by breathing and heartbeat.

As for the orientation of the subjects, we conducted corresponding supplementary experiments. Respiration and heartbeat signal of the subject were measured from different directions, and the results are shown in Fig. R5. The measurements were carried out in the subject's front, side and back orientations. It can be seen that accurate human respiration and heartbeat signals can still be obtained. It proves that the micro-movement caused by the respiration and heartbeat will be displayed in the entire human chest, that is, the chest wall, the chest side, and the back, so that the respiration and heartbeat signal of the human target can be captured from different angles.

Figure R5. Respiration and heartbeat signal of the subject measured from (a)(b) front, (c)(d) side, and (e)(f) back.

5. For the tracking, a video was watched on a screen. Is that a fair experiment? In the real-world, the drone would be moving in 3D, away from the tracking user, as well as in the horizontal plane.

Author response:

Thank you for your comment. The original tracking experiment is an off-line experiment, that is, the experimental verification of the UAV target track and the metasurface far-field measurement are not performed together. The original experiment has proven that the whole system is capable of responding to moving targets. However, it does not prove that the whole system has the ability to track the target in 3D space. Therefore, we have made a lot of improvements to the UAV tracking experiments in the revised manuscript. The specific improvements are as follows:

As mentioned above, the metasurface system was redesigned and has the capability of beam scanning for hemispherical coverage by simple reconfiguration of the code distribution. On this basis, UAV tracking in real world instead of video clips on display was carried out in the experiment. We preset a variety of 3D movement trajectories for the UAV, including radial movement away/closer from the subject and movement in the horizontal plane. The UAV flew on the preset trajectory and the observer located the object to be tracked in the eye tracker through rapidly blinking their eyes. The tracking algorithm then automatically tracked the selected target. The movement of the UAV was reflected on the eye tracker as a change in the viewing angle. The eye tracker-metasurface interface program then determined the EM response of metasurface and kept the EM beam turning to the angle corresponding to the UAV.

To prove that the system can track the target, the most intuitive way is to get information from the tracked target. In the present embodiment, the velocity of the UAV was collected. The experimental details and results are illustrated in Author response 6.

Different from the original off-line experiment, the on-line experiment is adopted in the revised manuscript. The on-line experiment not only allowed EM waves to irradiate the UAV, but also obtained the status information of the UAV.

6. In terms of the speed of tracking, the authors should discuss the speed of modulation of the metasurface. If it is much slower than the movement of the target, the target would already be away from the position by the time that the beam is steered to that angle.

Author response:

Thank you for your comment. Electrical modulation is characterized by extremely fast response speed. There are many scenarios that require real-time performance, such as military radar, which also use electrical modulation. In this paper, the modulation time of the metasurface is mainly composed of three parts: the switching time of PIN diodes, the switching time of steering-logic board and the running time of the tracking algorithm. For example, the PIN diodes (MADP-000907-14020) have a switching speed of 2-3 ns, which allows for operation at microwave and millimeter wave frequencies. The FPGA and DAC module operate at clock rates of up to MHz. The track algorithm maintains algorithm speed in the millisecond range. The above times are all in the order of milliseconds, microseconds or nanoseconds, that is, the modulation time of the metasurface is without a doubt very short, which guarantees the beam modulation speed.

In order to further improve the robustness as well as reduce the switching time, we

pre-select a series of EM wave radiation angles according to the far-field experiment results. Once the target is locked, the algorithm can track the UAV and provide its real-time position. When the UAV approaches the selected radiation angle, the EM wave will illuminate the corresponding angle in advance. The selected angle can be tuned depending on the application scenario. It should be noted that this example is to illustrate the design concept, implementation method and to prove that EM waves can illuminate the tracked target. Therefore, the velocity of the UAV was collected in the present embodiment. The principle of velocity measurement is as follows:

According to Doppler effect, when the moving target approaches the metasurface system, the frequency of the echo signal becomes higher than the frequency of the transmission signal; When the target is away from the radar, the echo signal frequency decreases. The difference between the frequency of the transmitted signal and echo signal is defined as the Doppler frequency:

$$f_d = \frac{2v_r}{\lambda} \quad (R3)$$

where v_r is the radial velocity of the target relative to metasurface, and λ is the wavelength of the emitted EM wave. It is generally agreed that f_d is negative when the target is far from the metasurface and positive when the target is close to the metasurface. It can be seen that the radial velocity v_r of the target can be obtained by the above formula if the Doppler frequency f_d can be measured when the emission wavelength λ is known. It can also be seen that the accuracy of velocity measurement mainly depends on the detection accuracy of the Doppler frequency f_d when the transmitting wavelength is fixed.

Suppose the CW signal emitted by the metasurface is:

$$u_t(t) = A_{tm} \cos(\omega_c t) \quad (\text{R4})$$

where A_{tm} is the amplitude of the transmitted EM signal, and ω_c is the angular frequency of the transmitted EM wave signal. The reflected echo signal by the target is denoted as:

$$u_r(t) = A_{rm} \cos(\omega_r t + \varphi) \quad (\text{R5})$$

where A_{rm} is the amplitude of the echo signal, ω_r is the angular frequency of the echo signal, and φ is the phase difference between the echo signal and the transmitted signal. By multiplying the echo signal and the transmitted signal, there arrives:

$$A_{tm} \cos(\omega_c t) \cdot A_{rm} \cos(\omega_r t + \varphi) = \frac{A_{tm} \cdot A_{rm}}{2} \{ [\cos(\omega_c + \omega_r)t + \varphi] + [\cos(\omega_r - \omega_c)t + \varphi] \} \quad (\text{R6})$$

In the formula, the first term is the sum frequency component (high frequency component), which can be filtered by the LPF low-pass filter; The second term is the difference frequency component (low frequency component), which can be retained by the low-pass filter. This difference frequency component is also known as the IF signal. The frequency of the IF signal is the Doppler frequency, which can be obtained by the FFT operation. Then the radial velocity of the target relative to the radar can be obtained by equation (R3). On this basis, the flying velocity of UAV can be derived according to the geometric relationship between radial velocity and flying velocity.

Herein, we take DJI mini 3 pro flying in an open area as an example. The experimental process was carried out in outdoor to mimic the application scenarios to

the greatest extent. The observer with normal vision worn eye tracking glasses which records eye movement data. The experiment results are depicted in Fig. R6, which illustrates the trajectory of the UAV, and compares the measured speed as well as reference speed of the UAV. It can be concluded that the proposed "glimpse-and-forget" metasurface smart system can track the moving target, which verifies the effectiveness of the design.

Figure R6. The flight trajectory, and reference speed when the UAV flies in the (a) vertical and (b) horizontal planes.

7. Overall, I am not sure what the tracking demonstration is showing that is more than the simple examples of beam steering control using the eye tracker in Figure 3.

Author response:

We have carefully considered the reviewer's comment and made revisions accordingly. As described in Author Response 5 and 6, our major revisions to "Glimpse-and-forget" metasurface smart target tracking system are as follows:

- extend the beam steering capability of the metasurface system from the horizontal plane to full space
- improve visible optical transmittance of metasurface structures by transparent dielectric substrates and metal mesh patterns
- measure the UAV in real world instead of on display
- collect the velocity of the UAV in real time

The experiment results prove that the moving target can be tracked in real time, which verifies the effectiveness of our design.

8. For speech recognition from microwaves, similar to an earlier comment, is the information still available if the speaker was facing the opposite way? We would be able to hear it with our ears, but does the metasurface need direct line of sight to detect the surface vibrations? If so, for a human's voice, would the metasurface have to be focused directly on the mouth?

Author response:

The author thanks the reviewer for the valuable comment. For respiration and heartbeat signal detection, if the human target faces away from the metasurface, we can still detect the respiration and heartbeat signal of the target. This is because the micro-movements of the body surface caused by the periodic breathing of the lung and the periodic beating of the heart can be reflected in the front wall of the chest, the left and right sides of the chest, and the back. Therefore, when the target is facing away from the metasurface, EM waves can still detect the target's breathing and heartbeat information.

However, speech detection is different. The vocal mechanism that produces speech is a very complex process, including the vibration of the Adam's apple, the movement of the tongue, the opening and closing of the mouth, and the movement of the cheek. The final speech signal is emitted from the front of the face and propagated into free space. Therefore, if the speaker is facing back to the metasurface, the voice information cannot be acquired in the EM echo. Metasurface requires a direct line of sight to detect surface vibrations. The focusing location should be Adam's apple, mouth, and cheeks, not just the mouth since the vibration of the mouth corresponds only to the low frequency component of the sound.

9. It would be interesting to hear the original voice and the recovered one from the reflected microwaves to compare them, as the waveform looks like it contains a lot of noise.

Author response:

Thanks to the reviewer's comments, we have attached the original voice signal and the voice signal after noise removal measured by metasurface as Supporting Materials. The reviewer can use any audio player to play the sound and compare them. Although the noise still exists, the enhanced speech result is enough to be recognized, and the recognition perfectly reconstructs the text information.

10. The authors should discuss the practicality of the platform that uses microwave frequencies with metasurfaces. The 40x50cm metasurface would be fairly large for a user to carry around. Additionally, the user would need an antenna along with other equipment in order to detect the reflected microwaves. Could this be integrated into a single metasurface that emits and receives microwave signals?

Author response:

Thanks for reviewer's comments. We have carefully considered the comments of the reviewers and made revisions accordingly. First, to address the issue of oversized metasurfaces, new metasurface design is adopted in the revised manuscript. In the new design, the phase modulation of the radiation-type metasurface unit cell is achieved through integrating two anti-symmetrically configured PIN diodes, thus achieving uniform EM radiation intensity but inverse radiation phase. By the modulation of the real-time coding distribution on the metasurface through the programmable bias circuit, the dynamic beam scanning capability is verified both numerically and experimentally. The completely new architecture simplifies and eliminates complex through-holes and vias. On this basis, we increase the operating frequency from 4 GHz

to 14.5 GHz, which also contributes to the reduction of metasurface size. Through the above improvements, the size of the metasurface is reduced from 50*50 cm² to 24*24 cm². The new architecture and design greatly simplify the system and improves the integration capability.

Next, traditional programmable metasurfaces based on reflective mode require external illuminating feed source, which inevitably complicates the system. To solve this problem, the metasurface architecture is redesigned without affecting the functionality of the system. The working mode of the metasurface is changed from reflection-type to radiation-type, and the transmitting antenna is removed in revised manuscript which greatly simplifies the metasurface structure, as illustrated in Author Response 1.

In the EM detection architecture of the work, EM waves irradiate the detected object, and then the state of the detected object can be obtained by analyzing the reflected wave. The metasurface undoubtedly has transmitting and receiving functions according to the far-field measurement results. When the metasurface is used for both transmitting and receiving, the detection can be regarded as a co-time co-frequency full-duplex (CCFD) reused metasurface process, that is, EM wave signals are simultaneously transmitted and received over the same frequency resource by one metasurface. To achieve this process, RF circulators should be used in the system, the VNA transmitter can be connected to port 1, while the metasurface is connected to port 2, and the VNA receiver is connected to port 3. When the transmitter power enters, it will be circulated to the metasurface, but not to the VNA receiver. On the other hand, when the metasurface

receives signals, they will be circulated to the VNA receiver and not the other way around. The receiver is isolated from the transmitter (about 20 to 25 dB), while the metasurface receives its power from the transmitter and allows the received signal to pass through to the receiver.

Figure R7. Block diagram of proposed RF interference cancellation technique.

However, it is not enough to use a circulator, because the isolation of the circulator is only about 20 to 25 dB, the signal received by the metasurface is still far less than the interference from the transmitter to the receiver. In other words, since CCFD transmits and receives signals at the same time and frequency, the transmitter would generate strong self-interference (SI) to the receiver. It is impossible to realize CCFD reused metasurface detection without developing anti-interference technology. To cancel the SI, the analog cancellation circuit tries to recreate a signal that matches the leaked interference signal for cancellation. The inverse of the transmitted signal is first obtained using a phase shifter with an attenuator. The attenuation and phase of the inverse signal are dynamically adjusted to match the self-interference leaking from the

RF circulator. After combining both inverse and leak signals, the received signal can be passed through the processing unit with the minimum effect of self-interference, as shown in Fig. R7. Some references are given below.

It is therefore possible to integrate the ability to transmit and receive microwave signals in a metasurface. Due to the direction of the research, we have only removed the transmitting antenna and given the expected scheme for removing the receiving antenna. If necessary, we would be happy to collaborate with CCFD-specialized researchers in the follow up.

[1] US Patent 5444864. <http://www.google.com/patents/US5444864>.

[2] US Patent 6539204. <http://www.google.com/patents/US6539204>.

[3] Phungamngern, N., Uthansakul, P. & Uthansakul, M. Digital and RF interference cancellation for single-channel full-duplex transceiver using a single antenna. *2013 ECTI-CON*, Krabi, Thailand, 212-216 (2013).

[4] Bharadia, D., McMilin, E., Katti, S. Full duplex radios. *ACM SIGCOMM*, New York, USA, 375-386 (2012).

[5] Hong, S., Mehlman, J., Katti, S. Picasso: Flexible RF and spectrum slicing. *ACM SIGCOMM*, New York, USA, 283-284 (2012).

[6] Hong, S., Mehlman, J., Katti, S. Picasso: Full duplex signal shaping to exploit fragmented spectrum. *ACM Workshop on Hot Topics in Networks*, New York, USA, 1-6 (2011).

11. Finally, the manuscript could do with an overall grammar check to improve the English presentation. The first word of the introduction needs an article to start with, and there are numerous other small mistakes that need to be cleaned up. Regarding Figure 4, the authors write ‘the black lines’, but they are blue, and there are various spelling mistakes such as ‘matasurface’ in the SI. It also seems that the radiation spectrum of Figure 3a is incorrect, it is going in the same direction as (b). Figures 5e and f also appear to be the same data.

Author response:

Thanks for reviewer’s comment. To correct the minor grammar errors and spelling

mistakes, we first read through the entire paper and checked the grammar several times. Then, we have asked a colleague who speaks English to correct the details in this manuscript. Finally, just in case, we used the polish service by of a professional team. We believe that the grammatical errors have been revised corrected and the readability as well as the logic of the manuscript have been greatly improved.

As for figure mistakes errors in the original manuscript, we have rechecked the figures again. This wase reason is due to negligence when in exporting the figures with using Origin 2019B software. In revised manuscript, we have fixed corrected these errors in the revised manuscript.

Reviewer #2 (Remarks to the Author):

(1) In this work, the eye movement is detected with a vision eye-tracking glass and then is used to control a dynamic metasurface for sensing the target. I notice that the similar idea has been explored in following two publications, i.e.,

<https://elight.springeropen.com/articles/10.1186/s43593-022-00019-x>

<https://elight.springeropen.com/articles/10.1186/s43593-022-00016-0>

In the above two publications, the brain signal is recognized using a specialized machine, and then to manipulate the metasurface for the purpose of sensing or communication. Although different signals (eye-movement sign, or brainwave signs) are considered, the underlying operational mechanism and procedures are almost the same for them. In addition, although a collection of experiments is presented, the trivial integration of vison and metasurface is just considered and the in-depth principle and

mechanism are lacking for the integration of eye (vision) and metasurface (microwave).

Author response:

Thanks for reviewer's comments. Due to their unique EM properties, metasurfaces have attracted great attention from engineers and researchers. In recent years, we have been exploring the significance of metasurface in social, medical, and humanistic care. With the help of eye tracking technology, we explore new advances in the human-machine interfaces, programmable metasurfaces and their interactions. This work proposes to capture multi-dimensional invisible information with the aid of human vision. In other words, this work realized the long-term human demand, to see what unseen. Moreover, we believe we have taken a major step forward in the metasurface intelligence by extending the EM application field. The corresponding multi-functional platform is able to realize the flexible radiation of EM beams and then auto-track moving targets. In addition, such vision-driven metasurfaces may expand the human senses and provide healthcare for the disabled. This is in accordance with one of the important future development directions of metasurfaces, that is, biologically-driven intelligent metasurfaces.

The innovations of two papers indicated by the reviewers lie in the exploration of a metasurface control method based on brain waves. Although physiological signals are combined in the two papers, physiological signals are only used as a control method. The proposed communication or remote-control function only transmits information or control signals, and there is no feedback mechanism in the system, that is, the EM waves sent out will not increase information obtained by human whose brain wave is

collected, let alone assist humans' decision-making.

Although this paper proposes a new way to achieve metasurface control using eye movements, it is not the only innovation in this paper. Different from the two papers above, the proposed system not only innovates the emission control method of EM waves, but also realizes a new environmental perception mechanism based on metasurface. Our paper focuses on perception enhancement based on metasurfaces, that is, improving human visual ability with the help of external information, so that people perceive information beyond optical band. Under the framework of this paper, the system realizes active detection based on human subjective control. That is, the system receives and analyzes echoes to obtain physiological signals, motion signals and voice signals from the environment, then provides more information perception and assists decision-making for human. As an example, a metasurface speech acquisition and enhancement system is proposed to overcome communication barriers for deaf people. By utilizing our scheme, clear and understandable complex voice sentence information instead of several simple instructions is retrieved which needs no further processing using sophisticated machine learning techniques. The system breaks through the barriers of visual information and auditory information, realizing the "visibility" of speech signals. This method undoubtedly improves the deaf people's ability to perceive the outside world, and also meets the future application needs of the disabled care. As another example, the system reduces the burden on human senses, allowing more tasks to be performed at the same time and increasing information processing ability. In addition, the proposed metasurface system realizes the multi-

functionality in a single platform without changing the physical configuration and expands the applicability of the system.

Figure R8. Metasurface system architecture listed in reference (a)[6], (b)[7], (c)[8] and (d)[9].

Our paper has its own development roadmap, not from brain waves. In recent years, computer vision has become one of the hot issues in the field of metamaterials and inspired many metasurface research^[1-3]. Computer vision enables computers and systems to derive meaningful information from digital images, videos and other visual inputs and take actions or make recommendations based on that information. Computer vision is used in industries ranging from energy and utilities to manufacturing and automotive and the market is continuing to grow. However, the development of computer vision also faces challenges, many scientific researchers want to understand better and imitate the biological vision system. In addition, more scientists have begun to systematically study new methods and mechanisms for eye movements, and the

development of eye movements in new fields has matured. Based on these studies, our paper builds an application bridge from eye tracking to computer vision, hoping to use human vision to explore the significance of metasurface in social, medical, and humanistic care.

It is well known that metasurfaces have been extensively investigated as artificially engineered planar sub-wavelength structures with versatile capabilities for delicate manipulations of EM waves. Since 2014, N. Engheta and T. J. Cui proposed the concepts of digital metasurfaces^[4] and coding metasurfaces^[5] in order to balance complexity and simplicity in the development in science and engineering. On this basis, EM metamaterials have developed their own research paradigm. A series of papers in similar system have been proposed and recognized by the academic community, widely distributed in holographic imaging, military applications, intelligent communications, network security and other fields ^[6-9]. Fig. R8 is the schematic diagram of references[6-9], which shows that our paper has a consistent research paradigm with these papers. Based on this research paradigm, our paper explores the principle and mechanism integration of vision and microwave information. Herein, we look forward to the future development of human intelligence, take human eye movement as the breakthrough point, focuses on the new intersection of human-machine interface and programmable metasurface, and explores the new progress and applications.

On the basis of manuscript, we have further improved in-depth integration degree of the system based on this research paradigm according to reviewer's comments. By using transparent dielectric substrates and metal mesh patterns, the proposed

metasurface structure can not only process microwave information, but also achieve high visible optical transmittance. It enables metasurfaces to integrate information in multiple frequency bands and the integration of eye (vision) and metasurface (microwave). Moreover, the reduced size of transparent metasurface can be compared to that of adult male face, which creates the prerequisites for all-in-one design of transparent glasses or masks. The human can observe the external environment through transparent metasurfaces, make basic decisions with the help of the observed optical information, and then tune EM waves to better detect the outside world. By analyzing echoes, microwave information can be further extracted, which can further assist human decision-making and enhance perception combined with optical information.

[1]Lu, H., Zhao, J., Zheng, B. et al. Eye accommodation-inspired neuro-metasurface focusing. *Nat. Commun.* **14**, 3301 (2023). <https://doi.org/10.1038/s41467-023-39070-8>

[2]Kogos, L.C., Li, Y., Liu, J. et al. Plasmonic ommatidia for lensless compound-eye vision. *Nat. Commun.* **11**, 1637 (2020). <https://doi.org/10.1038/s41467-020-15460-0>

[3]Li, W., Ma, Q., Liu, C. et al. Intelligent metasurface system for automatic tracking of moving targets and wireless communications based on computer vision. *Nat. Commun.* **14**, 989 (2023). <https://doi.org/10.1038/s41467-023-36645-3>

[4]Della Giovampaola, C., Engheta, N. Digital metamaterials. *Nat. Mater* **13**, 1115–1121 (2014). <https://doi.org/10.1038/nmat4082>

[5]Cui, T., Qi, M., Wan, X. et al. Coding metamaterials, digital metamaterials and programmable metamaterials. *Light Sci. Appl.* **3**, e218 (2014). <https://doi.org/10.1038/lsa.2014.99>

[6]Li, L., Jun Cui, T., Ji, W. et al. Electromagnetic reprogrammable coding-metasurface holograms. *Nat. Commun.* **8**, 197 (2017). <https://doi.org/10.1038/s41467-017-00164-9>

[7]Qian, C., Zheng, B., Shen, Y. et al. Deep-learning-enabled self-adaptive microwave cloak without human intervention. *Nat. Photonics* **14**, 383–390 (2020). <https://doi.org/10.1038/s41566-020-0604-2>

[8] Zhang, L., Chen, M.Z., Tang, W. et al. A wireless communication scheme based on space- and frequency-division multiplexing using digital metasurfaces. *Nat. Electron.* **4**, 218–227 (2021). <https://doi.org/10.1038/s41928-021-00554-4>

[9]Wei, M., Zhao, H., Galdi, V. et al. Metasurface-enabled smart wireless attacks at the physical layer. *Nat. Electron.* (2023). <https://doi.org/10.1038/s41928-023-01011-0>

(2) The details of eye tracking glasses used in this work are required for readers to

reproduce the reported results.

Author response:

Thanks for reviewer's comments. More details related on eye tracking glasses were added in revision. The principle and method of eye movement recognition were added in Methods Section and Supplementary Note S3. In addition, the brand and version information of the eye tracker (7invensun ASee glasses eye movement system) were added in the revision. The glass is capable of recording the user's observation behavior or operation process.

(3) Definitely, it needs to add more details (like, the time, power, accuracy, and so on) for the whole procedure (like Fig.7a), i.e., the recognition of eye movement, the control of metasurface, the perception of target, and the visualization of the target on the glasses.

Author response:

According to the reviewer's comment, more details related to the whole procedure were added in manuscript and Supplementary Information S10. Here, we take the section named "Barrier-free speech acquisition and enhancement system" as an example. The metasurface emitted a 14.5 GHz CW signal with 20 dBm power. The recording lasted for 10 seconds, with a sample rate of 5000 Hz, which satisfied the Nyquist sampling requirement for audio signals. Moreover, the audio files of original voice and the recovered voice were uploaded as attachments. On this basis, we have enriched the details in manuscript and Supplementary Information.

(4) As for the section of vision multi-targets' respiration and heartbeat detection, what is the maximal number of targets the proposed system can achieve? And, is there any constraints on the distance between the targets? In the presented results, only static targets are considered. How to deal with the moving targets?

Author response:

Thank you for your comment. On the basis of autonomous beam control following eye movements, the observer sequentially gazes at one of the human subjects in order to irradiate them with EM waves. When the person to be detected takes a breath, a portion of body parts, mainly the chest wall, moves periodically according to his breathing pattern. In addition, the mechanical deformation caused by heart beating would also be transmitted to the chest wall, superimposed on the micro-movements caused by respiration. The EM echoes reflected from the surface of the human body are modulated by these faint micro-movements of the chest wall. Since the system distinguishes the targets using EM beams following eye movements, the system's ability to discriminate between individuals depends on whether the processed echoes are loaded with only the respiratory and heartbeat information of a single subject. In other words, if when the processed echoes are loaded with only one subject's information, the system can distinguish that subject.

Here, we discuss the metasurface resolution, i.e., the ability of a metasurface to discriminate between two subjects. The resolution is defined as half the first-null beamwidth ($\text{FNBW}/2$). If two subjects are separated by an angular distance equal to or greater than $\text{FNBW}/2$, they can be resolved. According to the empirical formula, FNBW

is usually used to approximate twice the half-power beamwidth (HPBW)

$$\frac{FNBW}{2} \approx HPBW \quad (1)$$

HPBW is the angular separation in at which the magnitude of the radiation pattern decreases by 50% from the peak of the main beam. For uniform linear arrays, an approximation for HPBW is given as by the equation,

$$HPBW = \frac{0.886\lambda}{Nd \cos \theta} \quad (2)$$

Where d is the spacing between each unit cell, N is the number of elements in the radiation plane, λ is the operating wavelength and θ is the angle of the beam direction. Our horizontal area of focus (i.e., comfort zone) is 60° , while our vertical focal area of vision is 55° , as shown in Fig. R9. Within this field images are sharp, depth perception occurs. The imaging quality of the human eye deteriorates beyond this area, although eye movements are still present. Therefore, we focus on the situation where the metasurface radiation angles are within the comfort zone of the human eye. Taking into account the beam width, we set horizontal angle range of the metasurface EM beam from -28° to 28° and the maximum elevation angle to 25° . Although maximum horizontal and vertical EM radiation angle are provided according to eye movement, this does not mean that the proposed metasurface does not possess the capability of beam scanning for hemispherical coverage through simple reconfiguration of the code distribution.

In our demo, we set the condition to $\theta = 30^\circ$, $N = 16$, $d = 20.7$ mm and $d = 15$ mm, resulting in an HPBW of 5.1° (0.088rad). However, due to bias network on the top layer, machining and measurement error, the radiation directivity of the metasurface will

slightly decrease and the beam width will increase compared to the ideal situation. According to the simulated and experimental results, the HPBW is increased between 8° and 9.5° . Under these conditions, we can calculate the maximum number of targets. The range of EM radiation range of the system is 56° (from 28° to -28°), and the maximum number of people can be calculated by dividing the radiation range with the maximum HPBW. It can be concluded that the maximum number of targets the system can detect is at least 5.

Figure R9. Human eye (a) horizontal and (b) vertical fov.

For verification, the experiment was carried out to discriminate between multiple targets. Three volunteers were recruited to conduct the experiment, each wearing a piezoelectric respiratory belt to record the reference respiration signal. The azimuth angles at which the subjects were located were 0° , 12.5° and 28° . To avoid the interference of random body movements, the subjects under test were asked to remain seated in resting state. In each of the three sets of experiments, the observer fixated one of the subjects in turn in order to irradiate them with EM waves. As shown in Fig.R10, that respiration signals are detected in all three directions, which matched well with the

reference signals for both subjects. It proves that the system can distinguish the subjects in the above three directions. Due to the symmetry of the detection system, the system is able to clearly discriminate between at least five subjects. It is clear that the beamwidth of the metasurface has a large influence on the maximum number of targets detected.

Figure R10. Respiration detection results for the case where three subjects co-exists when observer fixate the azimuth of (a)0°, (b)12.5°, (c)28°.

For moving human targets, it is difficult to extract breathing and heartbeat signals of human targets from radar echoes, because the motion amplitude and frequency of limbs' displacement during walking are much larger than the body surface micro-movements caused by breathing and heartbeat. Therefore, breathing and heartbeat detection during walking is the most difficult point in radar-based life detection. Dr.

Changzhi Li wrote a commentary on this issue in nature electronics.^[1] For this problem, there is no feasible solution till now. In our follow-up research, we plan to focus on how to obtain the target breathing and heartbeat during walking.

[1] Li, C. Vital-sign monitoring on the go. *Nat. Electron.* **2**, 219–220 (2019). <https://doi.org/10.1038/s41928-019-0260-z>

(5) As for the “X-ray eyes”, it is not suitable since the eye has no ability of seeing through the obstacle. Maybe, the X-ray glasses is more suitable.

Author response:

Thanks for reviewer’s comments. In the revised manuscript, the expression "X-ray eyes" has been changed to "X-ray glasses". In addition, we have revised the description to emphasize that invisible microwave information is converted into visual information in revision. And then the microwave information after conversion can be seen by the human eye instead of original microwave signal.

(6) As for the section of barrier-free speech, I suggest to remove this part, since the metasurface here plays the communication relay, as intensively discussed in the area of

Author response:

We thank the reviewer for this comment. Metasurface has recently gained enormous attention from both academia and industry. As a type of relay, reconfigurable intelligent surface (RIS) is a two-dimensional surface made of artificial EM units controlled by integrated electronics. By changing the reflection characteristics of each unit cell, the RIS can proactively and intelligently adjust the wireless transmission

environment, flexibly configure the channel between transmitter and receiver and thus significantly improving the link quality of wireless communications. In the RIS relay system, the role of the metasurface is to control the path of communication information transmission through coding.

Figure R11. System architecture diagram (a) RIS as a communication relay and (b) "barrier-free speech acquisition and enhancement system".

In our work, metasurface first directs EM beam guided by eye tracker toward the loudspeaker to detect its surface vibrations driven by voice signal. The micro-displacement information of audio signal is carried in the phase of the echoes. The echoes of the target are measured and analyzed, thereby obtaining voice signals and the corresponding speech recognition results. In this way, sound wave information is

converted into microwave information, and then into visual signal directly visible to the human eyes.

For further illustration, we compare "barrier-free speech acquisition and enhancement system" with RIS as a type of relay, as shown in Fig. R11. It can be found that there are clear differences between the two in essence. The main difference is that the metasurfaces play different roles in the system. RIS acts as a relay in the link, providing low path loss and allowing EM waves to bypass obstacles. The EM waves emitted by the base station are transmitted to the user by reflection of RIS. In the whole process, RIS system realizes the information relay and does not change any information in the channel. For comparison, our system realizes the detection function, the EM waves emitted by the metasurface system are transmitted and reflected by the subject. The same system receives the echoes, then analyzes the modulated echo signal to extract the useful information, similar to conventional radar. During the whole process, there is no direct EM waves reflection on the metasurface. In other words, metasurface in the system is not relay.

Next, in the "barrier-free speech acquisition and enhancement system", metasurface is directly controlled by the user who needs to select the AOI, and then the metasurface begins to extract the selected target's information. In the RIS system, the user interacts with the terminal device, and the metasurface is not directly controlled by the user.

Thirdly, the two systems have different application scenarios. RIS is used in communication systems, while the "barrier-free speech acquisition and enhancement

system" has been used to provide healthcare access and overcome communication barriers for deaf people. The system is not intended for communication.

Therefore, the mechanism of speech detection is different from that of RIS communication, and the authors think it is better to keep this part. Since we did not express it clearly, we are sorry for your misunderstanding. We have modified the corresponding parts in the manuscript to make the expression more accurate and clearer.

(7) The figures need to be improved, because many legends and marker are blurred and not clear to me.

Author response:

We have carefully considered reviewer's comment and made revisions accordingly. In the revised manuscript, we have enhanced all the figures in manuscript and Supplementary Information. And the legends and markers in Fig. 1-5 in manuscript have been enlarged for clarity.

(8) It'd better if Fig. 1 and 2 are combined together.

Author response:

Thank you for your comment. In the revised manuscript, Figs. 1 and 2 were combined together as new Figure 1 in manuscript. Meanwhile, the caption and structure of the new Fig. 1 were readjusted.

(9) Why is the frequency (4.0GHz) chosen

Author response:

Thank you for your comment. We believe that microwave operating frequency has no significant impact on the principle of the detection system, and the principle of microwave detection is the same in different microwave frequency bands. The choice of operating frequency mainly depends on other parts of the system, such as the processing difficulty of metasurfaces, measurement conditions and other factors. That's why we chose 4.0 GHz in the original manuscript.

Following the reviewer's comments, we have made extensive modifications to the mechanism, design and functionality of the detection system. In the revised manuscript, we have increased the operating frequency to 14.5 GHz to accommodate new changes in metasurface mechanism, function and dimensions. The results show that the system still retains its original function.

Furthermore, shorter operating wavelengths have a better impact on the system. It significantly improves the displacement resolution and measurement accuracy of the metasurface. In addition, the increase in frequency can reduce the metasurface profile, which is of help to the miniaturization and integration of the system.

Reviewer #3 (Remarks to the Author):

The authors demonstrate respiration and heartbeat signal detection, subject detection located behind obstacle, and UAV tracking and so on. The proposed vision-driven metasurface technically sounds, but I could not see substantial improvement in metasurface design and its operating systems (concept of coding metasurfaces)

compared to many other prior arts.

Author response:

Figure R12. Metasurface structure listed in reference (a)[3], (b)[4], (c)[5], (d)[6,7] and (e)[8].

Thanks for reviewer's comments. The valuable comments from reviewers have made us take a closer look at our work. Metasurfaces, which are providers of unprecedented manipulations on EM waves upon interfaces, have become a popular topic in the academic community. Since 2014, N. Engheta and T. J. Cui proposed the concepts of digital metasurfaces^[1] and coding metasurfaces^[2]. By programming different coding sequences, a single digital metasurface has the ability to manipulate EM waves in different manners. The proposed coding metasurfaces, digital metasurfaces and programmable metasurfaces are very attractive for a variety of applications, such as

controlling the radiation beams, reducing the scattering features of targets and realizing other smart metasurfaces. A series of papers in similar system have been proposed and recognized by the academic community, widely distributed in holographic imaging, military applications, intelligent communications, network security, calculation and other fields^[3-7].

Figure R13. (a) Overall configuration of the proposed radiation-type metasurface with 16×16 units. Metasurface structure listed in reference (b) simulated and measured field patterns of metasurface at 14.5 GHz.

On this basis, we explore new advances in human-machine interfaces, i.e.

physiological information-metasurface interactions in the hope that it would have great significance in social, medical and humanistic care. The work expands the human senses in that not only the visible optical information can be acquired but also the invisible microwave information, in other words, this work realized the long-term human demand, to see what unseen. The proposed multi-functional metasurface system provides a large range of functions such as physiological information detection, position information detection, voice detection, target tracking and disabled health care.

We understand the comments made by the reviewers about metasurface design and concept. However, we want to emphasize that the innovation of this paper lies in the application value of metasurfaces in human-machine interfaces, i.e., physiological information-metasurface interactions. We believe our work meets the future application needs in social, medical and humanistic care. In addition, computer vision and human physiology have become hot issues in the field of metamaterials, and our work has established important connections between the two, inspiring future research. Therefore, I think our paper is of great value. In order to demonstrate that metasurface application innovation is still of great significance, we list the metasurface structural design proposed in references [3-8], as shown in Fig. R12. It can be seen that many excellent papers are still based on 1-bit or 2-bit coding metasurface design theory. For example, the structure proposed in the reference [6] is exactly the same as that in reference [7]. But it doesn't diminish the value of these papers in holographic imaging, military applications, intelligent communications, network security and other fields. On the contrary, these works have been widely recognized in the academic community and led

the development in above fields. As illustrated in reference [8], "our generic technique is not limited to implementations based on a specific programmable metasurface design."

Furthermore, we carefully considered the comments of the reviewers and made revisions in the metasurface design and integration. The improved metasurface structure and properties are shown in Fig.R13. The specific improvements are as follows: Firstly, we extend metasurface from one-dimensional to two-dimensional space. The metasurface possesses the beam scanning capability for hemispherical coverage through simple reconfiguration of the code distribution. Secondly, the metasurface's operating mode is changed from reflection-type to radiation-type. The transmitting antenna is removed, greatly simplifying the metasurface structure. Thirdly, by using transparent dielectric substrates and metal mesh patterns, metasurface can achieve high visible optical transmittance, which enables information integration in vision (light) and metasurface (microwave). Observers make basic decisions based on the observed optical information, and further extract more useful information in the microwave frequency band. Fourthly, a new design simplifies the metasurface structure, complex through-holes and vias are get rid of, metal patterns are etched only on top and bottom layers. Furthermore, the size of the metasurface is reduced from $50*50\text{ cm}^2$ to $24*24\text{ cm}^2$. The reduced size of transparent metasurface can be compared to that of human face, which creates the prerequisites for all-in-one design of transparent glasses or masks and enriches application scenarios.

We believe this work is in line with one of the important future development

directions of metasurfaces, namely bio-driven intelligent metasurfaces. This will lead to the development of the metasurface field and bring enough inspiration for other work. Moreover, the metasurface system uses a narrow beam to scan the space in front of the antenna, the human body target is located in the far field of the antenna, and the breathing and heartbeat of different targets are distinguished by different scanning beam angles. Therefore, this scheme overcomes the defect that the traditional wide beam coverage scheme separation algorithm may fail. Finally, thanks again for reviewer's valuable comments, which gives us a deeper understanding of metasurface and its development.

- [1] Della Giovampaola, C., Engheta, N. Digital metamaterials. *Nat. Mater* **13**, 1115–1121 (2014). <https://doi.org/10.1038/nmat4082>
- [2] Cui, T., Qi, M., Wan, X. et al. Coding metamaterials, digital metamaterials and programmable metamaterials. *Light Sci. Appl.* **3**, e218 (2014). <https://doi.org/10.1038/lsa.2014.99>
- [3] Qian, C., Zheng, B., Shen, Y. et al. Deep-learning-enabled self-adaptive microwave cloak without human intervention. *Nat. Photonics* **14**, 383–390 (2020). <https://doi.org/10.1038/s41566-020-0604-2>
- [4] Wei, M., Zhao, H., Galdi, V. et al. Metasurface-enabled smart wireless attacks at the physical layer. *Nat. Electron.* (2023). <https://doi.org/10.1038/s41928-023-01011-0>
- [5] Li, L., Jun Cui, T., Ji, W. et al. Electromagnetic reprogrammable coding-metasurface holograms. *Nat. Commun.* **8**, 197 (2017). <https://doi.org/10.1038/s41467-017-00164-9>
- [6] Zhang, L., Chen, M.Z., Tang, W. et al. A wireless communication scheme based on space- and frequency-division multiplexing using digital metasurfaces. *Nat. Electron.* **4**, 218–227 (2021). <https://doi.org/10.1038/s41928-021-00554-4>
- [7] Zhang, L., Chen, X. Q., Shao, R. W. et al. Breaking Reciprocity with Space-Time-Coding Digital Metasurfaces. *Adv. Mater.* **31**, 1904069 (2019). <https://doi.org/10.1002/adma.201904069>
- [8] Sol, J., Smith, D.R. & del Hougne, P. Meta-programmable analog differentiator. *Nat. Commun.* **13**, 1713 (2022). <https://doi.org/10.1038/s41467-022-29354-w>

REVIEWER COMMENTS

Reviewer #1 (Remarks to the Author):

The authors answered the questions sufficiently. At least, now I endorse the publication of the manuscript with the consideration of other reviewers' comments and the judgement of editor.

Reviewer #2 (Remarks to the Author):

Report on NCOMMS-22-44752B

Authors have partially addressed my previous concerns; however, several important issues remain unclear to me.

1. In this work, the eye movement needs to be detected using near-infrared light source to illuminate the center of the eyes. In this scenario, I am afraid that such operation has some ethical issues in experiments and applications.
2. As all reviewers mentioned in the last round, in principle the presented strategy is very similar to the eLight paper. In this sense, the novelty is not clear.
3. For the UVA's results, it is more crucial to determine the accurate locations of UVA rather than speed alone. Thus, it is critical to examine the localization accuracy.
4. For the setting of 'speech acquisition and enhancement system', the loudspeaker, when it works, has usually notable vibration, and modulate the illuminated microwave signal. However, for human speech, the related vibration is really non-noticeable. In this case, how can we detect the tiny Doppler effect? Or, what kind of conditions we need for such detection?
5. Some additional references are listed in the response letter. Are they added in main text?

Reviewer #3 (Remarks to the Author):

The authors improved the manuscript quality according to reviewers' comments and suggestions. Extensive additional experimental demonstrations like fullspace beam scanning and smart target tracking systems are appreciated. Now I would like to recommend the acceptance of the paper in this journal.

Reviewer #2 (Remarks to the Author):

1. In this work, the eye movement needs to be detected using near-infrared light source to illuminate the center of the eyes. In this scenario, I am afraid that such operation has some ethical issues in experiments and applications.

Author response:

Thanks for reviewer's comments. The eye movement system (7invensun Asee glasses) is compliance with Low Voltage Directive (LVD) EN 62471:2008. The LVD aims to ensure that electrical equipment on the market fulfils the requirements providing for a high level of protection of health and safety of persons and property. It gives guidance for evaluating the photobiological safety of lamps and lamp systems including luminaires. Specifically, it specifies the exposure limits, reference measurement technique and classification scheme for the evaluation and control of photobiological hazards from all electrically powered incoherent broadband sources of optical radiation. We have uploaded the LVD report of the eye tracking system as an attachment. The report proves that the detection does not cause physical harm to the eyes, retina, vision and skin. Therefore, it can be concluded that the experiment is safe to the subjects. In addition, the subjects involved in this study are all informed about the research content, methods, and purposes. The Consent Form of all the experiments have been uploaded as an attachment. Moreover, we have added the following statements to the Supplementary Information" The eye movement system (7invensun Asee glasses) is compliance with Low Voltage Directive (LVD) EN 62471:2008, which proves that the detection does not cause physical harm to the eyes, retina, vision and

skin."

2.As all reviewers mentioned in the last round, in principle the presented strategy is very similar to the eLight paper. In this sense, the novelty is not clear.

Author response:

Thanks for reviewer's comments. In this paper, we expand the human senses to acquire not only visible optical information, but also the invisible microwave information. In other words, this work realized the long-term human demand, to see what unseen. Focusing on the demand for new human development modes such as Metaverse/VR/AR, on the eve of the explosion of consumer products, we provide a feasible "X-ray" glasses solution in the hope that it would have great significance in social, medical and humanistic care. One review paper evaluates our work "This opens up a wide range of possibilities, from providing real-time information and navigation guidance to immersive gaming experiences, all directly projected onto the user's vision."¹ It is clear that our work originates from human vision, obtains information from the outside world in real time by means of eye movement, and mainly develops applications such as breathing and heartbeat detection, voice detection, target tracking, and optically invisible position/action detection. In contrast, main applications on eLight paper are communication and remote control, which are completely different from our work.

Figure R1. The information flow of (a) this work and (b) eLight papers.

Next, in this work, optical information about the external environment can be received by the observer through transparent metasurfaces, which then help the observer make basic decisions about how to detect the outside world through EM waves. By analyzing echoes, microwave information can be extracted, which can further assist human secondary decision-making and enhance perception combined with optical information. We organically combine human vision with the external environment, which has both human active control and real-time analysis of the external environment. The optical information from the external world and the microwave information

obtained from real-time analysis constantly affect human decision-making. In the system, the entire information flow is closed-loop. In contrast, the work processes of the eLight papers are brain waves-EM waves-communication/control signals. The information flow is one-way instead of closed loop, the EM waves do not return information after they are emitted. If information needs to be returned, multiple metasurface systems are required. Therefore, it can be seen that the information flow in our work is different from the eLight papers, which also explains the difference in the paper architecture.

Thirdly, based on 2D EM modulation, our paper involves applications including breathing and heartbeat detection, voice detection, and optically invisible position/action detection. Those applications involve information characterizes as micro-Doppler signature that can be utilized to discriminate different human states. By applying short-time Fourier transform based time-frequency analysis to the collected scattered EM echoes, the human states including vital signs, voices, locations as well as working information can be obtained. For target tracking, we have developed speed and tracking algorithms. The two papers on eLight, one deals with communication technologies, such as encoding and decoding; the other article does not deal with any application technology.

Fourthly, the proposed metasurface architecture is totally different. The proposed radiation-type metasurface gets rid of transmitting antenna and greatly simplifies the system structure, which lays the foundation for the deep integration of "X-ray" glasses-metasurface system. In contrast, the metasurface proposed on the eLight is reflective-

type, it is unable to remove the transmitting horn in the metasurface system. It makes its EM emission system slightly more complex. More importantly, thanks to the reviewers' suggestions, the new designed metasurface structure can take into account the transmission of light when emitting microwaves, that is, it enables metasurfaces to integrate information in multiple frequency bands and the integration of eye (vision) and metasurface (microwave). The human can observe the external environment through transparent metasurfaces and then tune EM waves to better detect the outside world, which allows for a tighter integration of our metasurface system. It starts from the visual properties of the human eye and creates the prerequisites for transparent masks or eyeglass. In contrast, the metasurfaces proposed in the eLight papers operate only in the microwave band, and the microwaves emitted by the metasurfaces do not have a deeper integration process with the brain waves.

Figure R2. Digital and coding metasurface knowledge tree.

Table R1 The comparison between our work and eLight Papers

	Our work	Two eLight Papers	
Applications	Perception Enhancement	Communication	Remote Control
Experimental Project	Vital Signs Detection, Voice Detection, Target Tracking, Optically Invisible Position & Motion Detection	Text Communications	/
Technologies	Micro Doppler, Tracking Algorithm	Encoding & Decoding	/
Information flow	Loop	One-way	
Metasurface Emission	Radiation	Reflection	
Operation Band	Microwave & Light	Microwave	
Transmitting Antennas	No	Yes	
Optical transparency	Yes	No	
Environmental Information Acquisition	Yes	No	
Invisible Information Acquisition	Yes	No	
Peripherals	Eye Tracker	EEG cap and apply conductive paste	
Preparation/User Operation	Simple	Complex	
Potential metasurface & peripheral integration approaches	Transparent metasurface masks & eyeglass	/	

Furthermore, N. Engheta² and T. J. Cui³ developed digital and coding metasurfaces in order to balance complexity and simplicity in the development in science and engineering since 2014. As shown in Fig.R2, the research paradigm, that is, informational design-informational analysis-informational processing, has been recognized by the academic community, widely used in cloak⁴⁻⁵, holograms⁶, communications⁷⁻⁹ and others fields, including our work and the work on eLight. Therefore, our work follows a widely accepted research paradigm, and does not come

from a few papers. Our work is totally different from the works on eLight in terms of structures, technologies, applications, information flow, etc. Even for metasurface control, eye tracking and brain-computer interface belong to different physiological systems. Both of them have thousands of research achievements in their respective fields. They have significant differences in terms of measurement objects, signal sources, application domains, and signal analysis. In terms of application, the signal-to-noise ratio for electroencephalogram is too low, and it is difficult to extract the details we want. Moreover, it is more troublesome to collect, and it is necessary to apply conductive paste. Compared with the electroencephalogram, the eye tracking application may be closer to the actual landing.

Figure R3. Metasurface system architecture in reference (a)[7], (b)[8], (c)[9] and metasurface structures in (d)[8] and [10].

Many papers are not innovative in all aspects, but they are still of great value. As shown in Fig. R3, a series of RIS papers⁷⁻⁹ have same control methods, applications, and information flow, their differences are technical improvements; another example is that some works use the exact same metasurface structure and control method^{8,10}, but

they apply the same metasurface to a completely new field. These works have been widely recognized. Our work has made progress in terms of structure, control method, application, information flow, technology, etc., so we believe our work is innovative. Finally, we detail the differences between our work and the two papers published on eLight, as shown in Table R1.

- [1] Shaker, L.M., Al-Amiery, A., Isahak, W.N.R.W. *et al.* Metasurface contact lenses: a futuristic leap in vision enhancement. *J Opt* (2023). <https://doi.org/10.1007/s12596-023-01322-7>
- [2] Della Giovampaola, C., Engheta, N. Digital metamaterials. *Nat. Mater* **13**, 1115–1121 (2014).
- [3] Cui, T. J., Qi, M. Q., Wan, X., Zhao, J. & Cheng, Q. Coding metamaterials, digital metamaterials and programmable metamaterials. *Light Sci. Appl.* **3**, e218 (2014).
- [4] Yang, Y. *et al.* Full-polarization 3D metasurface cloak with preserved amplitude and phase. *Adv. Mater.* **28**, 6866–6871 (2016).
- [5] Qian, C., Zheng, B., Shen, Y. *et al.* Deep-learning-enabled self-adaptive microwave cloak without human intervention. *Nat. Photonics* **14**, 383–390 (2020).
- [6] Li, L., Jun Cui, T., Ji, W. *et al.* Electromagnetic reprogrammable coding-metasurface holograms. *Nat. Commun.* **8**, 197 (2017).
- [7] Wang, S.R., Dai, J.Y., Zhou, Q.Y. *et al.* Manipulations of multi-frequency waves and signals via multi-partition asynchronous space-time-coding digital metasurface. *Nat. Commun.* **14**, 5377 (2023).
- [8] Zhang, L., Chen, M.Z., Tang, W. *et al.* A wireless communication scheme based on space- and frequency-division multiplexing using digital metasurfaces. *Nat. Electron.* **4**, 218–227 (2021).
- [9] Ke, J.C., Dai, J.Y., Zhang, J.W. *et al.* Frequency-modulated continuous waves controlled by space-time-coding metasurface with nonlinearly periodic phases. *Light Sci Appl* **11**, 273 (2022).
- [10] Zhang, L., Chen, X. Q., Shao, R. W. *et al.* Breaking Reciprocity with Space-Time-Coding Digital Metasurfaces. *Adv. Mater.* **31**, 1904069 (2019).

3. For the UVA's results, it is more curial to determine the accurate locations of UVA rather than speed alone. Thus, it is critical to examine the localization accuracy.

Author response:

Thanks for reviewer's comments. The purpose of the "glimpse-and-forget" metasurface smart system is not only to measure the velocity of the UAV, but to prove that EM waves can irradiate and then track it. On this basis, we can track UAVs and jam them with the help of microwave in the outdoor environment. Two intentional

vulnerabilities of UAVs including jamming and navigation spoofing, can be exploited in this way. EM waves incident to UAVs can carry high-power noise or counterfeit navigation signals in order to hinder a navigation service or produce fake navigation positions. In the revised manuscript, we have added this expression to the main text.

The tracking procedure is denoted as follows: First, the observer locates the object to be tracked through blinking eyes quickly. Next, the tracking algorithm tracks the object and obtains its position in real time. Finally, digital-coding sequences of the programmable metasurface are changed to keep the radiation beam turning to the angle corresponding to the tracked object. Therefore, in the architecture of the paper, the localization of the UAV is obtained in real time through optical information. The angle of the UAV can be obtained directly from the image captured by the eye tracker, while the distance measurement of the UAV can be seen as monocular vision ranging, that is, using a digital camera to capture a single image to measure distance.

The basic principle of monocular visual ranging is as follows: first, a definite or empirically estimated length should be found, and then the target distance is estimated from the geometric relationship between the target and the known length in the image, as shown in Fig. R4. The origin of the camera coordinate system is the light center O point. The distance between the origin of the camera coordinate system and the origin of the image coordinate system can be expressed as the focal length f of the lens. P is the target under test and P' is the imaging point. x' is the imaging size and x_c is the known or empirically estimated target size. Ideally, the relationship between focal length f and target size and image size, as well as target distance z when the target is clearly imaged can be obtained by combining the figure with the triangle similarity principle as shown in Equation R1,

$$z = \frac{x_c}{x'} f \quad (\text{R1})$$

Figure R4. Imaging principle of pinhole imaging model.

The above is the simplest scenario for monocular ranging. In recent years, the monocular ranging has been widely studied, and various algorithms for different application environments have been developed. Some references are given below¹⁻⁶ including UAV ranging method based on monocular vision. Due to the direction of the research, we only give the preliminary scheme to determine the location of the UAV. If necessary, we would be happy to collaborate with researchers specializing in image processing in the follow up.

Finally, I would like to discuss the limitations of metasurface microwave localization measurement, in order to explain why we use optical methods to measure the localization of the target instead of microwave method in our design architecture. Different from traditional radar, since each metasurface unit cell responds differently to the broadband wave signal, neither the radiative metasurface proposed in this paper nor the reflective metasurface can form a stable broadband signal waveform in the EM irradiation region, resulting in the inability to achieve target ranging. For further illustration, we construct a metasurface schematic diagram, as shown in Fig. R5.

Assuming that the incident broadband FMCW signal is expressed as $s(t)$. Then the signal modulated by each unit can be expressed as,

$$S(\theta, \phi, t) = \sum_{i=1}^M \sum_{j=1}^N s(t) \cdot \Gamma_{i,j}(f)|_{f=Kt} \cdot e^{j\Phi_{i,j}(\theta, \phi)} \cdot e^{-jkR_{i,j}} \quad (\text{R2})$$

where i and j represent the number of unit cells in the x and y directions, respectively. K represents the frequency modulation slope of the FMCW signal, with $f = Kt$. $\Gamma_{i,j} = e^{j\psi_{i,j}(f)}$ represents the phase response of (i,j) -th unit, where $\psi_{i,j}(f) \in \{\psi_1(f), \psi_2(f)\}|_{\psi_2(f) - \psi_1(f) \approx 180^\circ}$. $R_{i,j}$ is the feeding distance from the EM feed point to each unit, which reflects the phase difference from the feed point to the unit cell. $\Phi_{i,j}(\theta, \phi) = e^{jk(i-1)d_x \sin\theta \sin\phi + jk(j-1)d_y \sin\theta \cos\phi}$ is the steering vector, in which d_x and d_y are the unit length along the x and y directions, respectively. For the spatial angle (θ, ϕ) , the relative geometric relationship and the unit coding state are determined. Then we can get the following expression,

$$S(t)|_{(\theta, \phi)} = s(t) \sum_{i=1}^M \sum_{j=1}^N e^{j\psi_{i,j}(f)}|_{f=Kt} \cdot e^{j\Phi_{i,j}} \cdot e^{-jkR_{i,j}} = s(t) \cdot e^{j\varphi(t)} \quad (\text{R3})$$

where $\varphi(t)|_{(\theta, \phi)} = \sum_{i,j}(\psi_{i,j}(t) + \Phi_{i,j} - kR_{i,j})$. It is clear that the additional modulation related to the angle and time has introduced to original broadband FMCW signal. The waveforms changes in each direction and at each moment. For Traditional ranging process, the transmitted signal is treated as the reference signal for matched filtering processing. However, the broadband emission signal is different at each spatial angle for metasurface. Therefore, mismatch problems will happen in matched filtering process, resulting in the inability to ranging. We believe that with the rapid development of metasurfaces and signal processing technology, the problem of non-collaborative target ranging will be solved in the near future.

Figure R5. Schematic diagram of a radiation metasurface.

- [1] Godard, C., Aodha, O. M., Brostow, G. J. Unsupervised Monocular Depth Estimation with Left-Right Consistency. *2017 IEEE Conference on Computer Vision and Pattern Recognition (CVPR)*. 6602-6611 (2017).
- [2] Raza, M., Chen, Z., Rehman, S. U. *et al.* Framework for estimating distance and dimension attributes of pedestrians in real-time environments using monocular camera. *Neurocomputing* **275**, 533-545 (2018).
- [3] Fernández Llorca, D., Hernández Martínez, A., García Daza, I. Vision-based vehicle speed estimation: A survey. *IET Intell. Transp. Syst.* **15**, 987–1005 (2021). <https://doi.org/10.1049/itr2.12079>
- [4] Wang, X., Zeng, P., Cao, Z. *et al.* (2023). A Monocular Vision Ranging Method Related to Neural Networks. (2023) https://doi.org/10.1007/978-3-031-36819-6_8
- [5] Choi, H., Kim, Y. UAV guidance using a monocular-vision sensor for aerial target tracking. *Control Eng. Pract.* **22**, 10-19 (2014).
- [6] Rong, Y., Zhang, Z., Qiu, C. *et al.* UAV ranging method based on monocular vision. *J. Phys.: Conf. Ser.* **1873**, 012041 (2021).

4. For the setting of ‘speech acquisition and enhancement system’, the loudspeaker, when it works, has usually notable vibration, and modulate the illuminated microwave signal. However, for human speech, the related vibration is really non-noticeable. In this case, how can we detect the tiny Doppler effect? Or, what kind of conditions we need for such detection?

Author response:

The authors thank the reviewers for their comments. The human vocal mechanism is a complex process, involving the participation of multiple organ parts, such as the vibration of the Adam's apple, tongue movement, mouth opening and closing, cheek oscillation, etc. The vibration of the Adam's apple generates a fundamental frequency speech signal. Through the movement of the tongue and the adjustment of the mouth and cheeks, a complete longitudinal sound wave signal is formed and transmitted into the air. Therefore, when measuring human speech signals with EM waves, only very low-frequency modulation envelope signals can be collected if EM waves focus on a person's cheek and mouth. It is common practice to focus the EM wave onto the human body's throat and collect relatively high-frequency signals from the throat and nearby skin and muscles.

Figure R6. (a) Waveform of raw recorded speech. (b) Waveform of processed speech signal.

As verification, an experiment was carried out for human voice acquisition. The EM beam generated by metasurface irradiated subject's throat. The subject read aloud the following words, "Consider Support Fit Hang Protect Professional." The signal

reflected by the throat was collected by VNA and then analyzed by signal processing algorithm proposed in this paper. Fig.R6 (a) denotes the waveform of voice file collected by the recording software in the field environment. The voice data processed by the algorithm is illustrated in Fig. R6 (b). Moreover, we have attached the original voice signal and the voice signal processed by the algorithm as Supporting Materials.

By comparing waveforms and sounds, it is clear that the system with an operating frequency of 14.5 GHz can detect voice signals. The collected signal from human is vaguer than that from loudspeakers, because the vibration of loudspeakers is more obvious than that of people's organs, as the reviewer pointed out. In order to obtain better voice detection results, one way is to increase the operating frequency to millimeter wave band or terahertz band, shorter operating wavelengths contribute to more precise micro-motion extraction. Moreover, since the beamwidth is proportional to the operating wavelength and inversely proportional to the metasurface size, larger metasurface sizes and smaller wavelengths reduce the beamwidth, allowing more energy to be focused on the desired area, improving the signal-to-noise ratio, and then improve the voice detection effect.

5. Some additional references are listed in the response letter. Are they added in main text?

Author response:

Thanks for reviewer's comments. The listed references are those mentioned in all versions of the response letter. In the revised manuscript, we reorganize the references and add these references to the main text as references [30]-[31], [33]-[45] and [55].

Moreover, the following text has been added in the revised manuscript, "since 2014, N. Engheta and T. J. Cui proposed the concepts of digital metasurfaces and coding metasurfaces in order to balance complexity and simplicity in the development in science and engineering. On this basis, EM metamaterials have developed their own research paradigm. Many exotic functionalities can be realized such as cloak, computer vision, holograms, communications, brain waves and others. " Finally, I would like to thank the reviewers and editors again, we have gained a lot during the whole revision process, and our understanding of EM field has been deepened.

Added References

- [30] Della Giovampaola, C., Engheta, N. Digital metamaterials. *Nat. Mater* **13**, 1115–1121 (2014).
- [31] Cui, T. J., Qi, M. Q., Wan, X., Zhao, J. & Cheng, Q. Coding metamaterials, digital metamaterials and programmable metamaterials. *Light Sci. Appl.* **3**, e218 (2014).
- [33] Qian, C., Zheng, B., Shen, Y. et al. Deep-learning-enabled self-adaptive microwave cloak without human intervention. *Nat. Photonics* **14**, 383–390 (2020).
- [34] Lu, H., Zhao, J., Zheng, B. et al. Eye accommodation-inspired neuro-metasurface focusing. *Nat. Commun.* **14**, 3301 (2023).
- [35] Kogos, L.C., Li, Y., Liu, J. et al. Plasmonic ommatidia for lensless compound-eye vision. *Nat. Commun.* **11**, 1637 (2020).
- [36] Li, L., Jun Cui, T., Ji, W. et al. Electromagnetic reprogrammable coding-metasurface holograms. *Nat. Commun.* **8**, 197 (2017).
- [37] Zhang, L., Chen, M.Z., Tang, W. et al. A wireless communication scheme based on space- and frequency-division multiplexing using digital metasurfaces. *Nat. Electron.* **4**, 218–227 (2021).
- [38] Ke, J.C., Dai, J.Y., Zhang, J.W. et al. Frequency-modulated continuous waves controlled by space-time-coding metasurface with nonlinearly periodic phases. *Light Sci Appl* **11**, 273 (2022).
- [39] Wang, S.R., Dai, J.Y., Zhou, Q.Y. et al. Manipulations of multi-frequency waves and signals via multi-partition asynchronous space-time-coding digital metasurface. *Nat. Commun.* **14**, 5377 (2023).
- [40] Ma, Q., Gao, W., Xiao, Q. et al. Directly wireless communication of human minds via non-invasive brain-computer-metasurface platform. *eLight* **2**, 11 (2022).
- [41] Zhu, R., Wang, J., Qiu, T. et al. Remotely mind-controlled metasurface via brainwaves. *eLight* **2**, 10 (2022).
- [42] Wei, M., Zhao, H., Galdi, V. et al. Metasurface-enabled smart wireless attacks at the physical layer. *Nat. Electron.* (2023).
- [43] Zhang, L., Chen, X. Q., Shao, R. W. et al. Breaking Reciprocity with Space-Time-Coding Digital Metasurfaces. *Adv. Mater.* **31**, 1904069 (2019).
- [44] Sol, J., Smith, D.R. & del Hougne, P. Meta-programmable analog differentiator. *Nat. Commun.* **13**, 1713 (2022).
- [45] Li, W., Ma, Q., Liu, C. et al. Intelligent metasurface system for automatic tracking of moving targets and wireless

- communications based on computer vision. *Nat. Commun.* **14**, 989 (2023).
- [51] Li, C. Vital-sign monitoring on the go. *Nat. Electron.* **2**, 219–220 (2019).

REVIEWERS' COMMENTS

Reviewer #2 (Remarks to the Author):

Authors have addressed my concerns well, thus I am happy to recommend its acceptance as it is.
Thanks for authors' efforts.